# Timing anthropogenic stressors to mitigate their impact on marine ecosystem resilience

Paul Pao-Yen Wu [1,2], Kerrie Mengersen[1,2], Kathryn McMahon[3,4], Gary A. Kendrick [4,5], Kathryn Chartrand[6], Paul H. York [6], Michael A. Rasheed[6] & M. Julian Caley[1,2]

Better mitigation of anthropogenic stressors on marine ecosystems is urgently needed to address increasing biodiversity losses worldwide. We explore opportunities for stressor mitigation using whole-of-systems modelling of ecological resilience, accounting for complex interactions between stressors, their timing and duration, background environmental conditions and biological processes. We then search for ecological windows, times when stressors minimally impact ecological resilience, defined here as risk, recovery and resistance. We show for 28 globally distributed seagrass meadows that stressor scheduling that exploits ecological windows for dredging campaigns can achieve up to a fourfold reduction in recovery time and 35% reduction in extinction risk. Although the timing and length of windows vary among sites to some degree, global trends indicate favourable windows in autumn and winter. Our results demonstrate that resilience is dynamic with respect to space, time and stressors, varying most strongly with: (i) the life history of the seagrass genus and (ii) the duration and timing of the impacting stress.

[1] Australian Research Council Centre of Excellence in Mathematical and Statistical Frontiers, University of Melbourne, Melbourne, VIC 3010, Australia. [2] School of Mathematical Sciences, Queensland University of Technology, GPO Box 2434, 2 George Street, Brisbane, QLD 4001, Australia. [3] School of Sciences and Centre for Marine Ecosystems Research, Edith Cowan University, 270 Joondalup Drive, Joondalup, WA 6027, Australia. [4] WAMSI Headquarters, M095, University of Western Australia, 35 Stirling Highway, Crawley, WA 6009, Australia. [5] UWA Oceans Institute and School of Biological Sciences, University of Western Australia, 35 Stirling Highway, Crawley, WA 6009, Australia. [6] Centre for Tropical Water & Aquatic Ecosystem Research, James Cook University, PO Box 6811, 14-88 McGregor Road, Cairns, QLD 4870, Australia. Correspondence and requests for materials should be addressed to P.P.-Y.W. (email: p.wu@qut.edu.au)

Anthropogenic stressors are degrading valuable marine ecosystems worldwide, particularly in coastal areas with high levels of human development[1–3]. Corals[4], seagrasses[5,6] and mangroves[7] are all rapidly declining globally, at least in part due to stressors associated with water quality, including light reduction, exposure of toxins and smothering by sediments. However, currently available approaches for mitigating the effects of these stressors are often limited by a poor understanding of how anthropogenic stressors and natural disturbance interact to influence myriad ecological processes. Such complex interactions can produce non-linear, additive and synergistic cumulative responses[8].

A key measure of impact captured by the emergent response of an ecosystem to a stressor is resilience[8,9]. As resilience is an emergent property of a system under this definition, it is underpinned by concepts of time, baseline and alternate processes and structures, and sets of metrics and criteria to quantify such processes and structures. However, tools for quantitatively evaluating resilience under changing ecological baselines remain a key challenge[10]. Here, we model complex ecological interactions to quantify resilience to an impact using three widely applied criteria: (1) resistance[8,11], the loss of individuals and/or species as the result of stress, (2) recovery[12], the expected recovery time and (3) persistence[9] or risk of local extinction (probability of zero population of a species) following stress. We focus on ecological resilience[10] as first proposed by Holling[9] rather than engineering resilience which focuses exclusively on recovery[10]. The former is a broader definition centred around the set of processes and structures describing an ecosystem and is widely applied in ecology[10].

Resilience in response to a stressor could be improved by taking advantage of ecological windows, periods planned in advance during which a specific stressor can occur with minimal impact on an organism or ecosystem[13]. Windows have been used to manage anthropogenic activities such as dredging[14]. Ecological windows differ from environmental windows detailed in existing regulatory frameworks (eg, U.S.A. National Environmental Policy Act 1969), which typically do not consider site-specific biological, environmental and stressor interactions[14]. Ecological windows also differ from windows of opportunity[15,16], which have a broader socio-ecological focus where typically unplanned events can trigger opportunities for wide-scale institutional and ecological changes.

Ecological windows are potentially useful for managing the ecological impact of anthropogenic disturbances and resilience[17–19], and opportunities to customise their application to local conditions are plentiful[14]. Indeed, their application in these management contexts may be preferable to the application of environmental windows. The latter can lead to quite simple restrictions where anthropogenic activities likely to impact a given ecosystem are prohibited during a time when a critical biological function is thought to occur. In addition, they can ignore processes that are equally important in conferring resilience that occur around these potentially critical times. Examples of this sort of restriction include restrictions on dredging during coral spawning in Western Australia[20] and restrictions on dredging during Pacific herring spawning in San Francisco Bay[14]. In contrast, ecological windows can emerge from complex interactions between disturbances and ecosystems making data requirements for their analysis onerous. Consequently, data-based quantitative estimation of windows has proven difficult[14].

To better understand resilience and its management using ecological windows, we developed a whole-of-system dynamic Bayesian network (DBN) model of an ecosystem and the stressors to be managed. We have adopted a modelling approach here because it would be impossible to estimate impact experimentally with sufficient certainty given the complex, variable and uncertain nature of ecosystems. The DBN approach integrates expert knowledge and available data using the established ecological windows framework to estimate the timing and length of windows for specific locations and time periods. Our DBN model presents an opportunity to predict the emergent response and resilience of a system given temporal variation in baseline environmental and biological conditions and their interaction with different timing, duration and magnitude of stressors.

Specifically, we study the impact of scheduled dredging and its associated stressors on the resilience of seagrass meadows as a canonical example of ecological windows. Dredging is a major source of disturbance affecting water quality with hundreds of millions of cubic metres of sediment dredged annually, most of which is associated with coastal development activities including port expansion and maintenance, land reclamation, coastal construction and shoreline protection, and offshore energy exploration[21]. This dredging can impact primary producer ecosystems such as corals, seagrasses and mangroves, especially through stressors such as light reduction and burial by sediments[22]. Although dredging produces multiple disturbances including impacts on pH and dissolved oxygen, the predominant stress on seagrass is light reduction arising from suspended sediments[13,22]. Periods and levels of light reduction emerge as key variables from complex interactions between spatial and temporal factors including the mechanics of the dredge, relative location of dredging to a seagrass meadow, wind and waves, storms, tides and associated water circulation[22]. As a result, impact assessments of dredging campaigns need to be customised for specific meadows at specific periods in time, and incorporate uncertainty associated with forecasted future conditions.

To better understand the utility of ecological windows, we modelled the responses of 28 seagrass sites distributed worldwide that include examples of the three main life history types expressed by seagrass genera. We focused on colonising *Halophila*, opportunistic *Zostera* and persistent *Amphibolis* seagrass genera[23]. *Halophila* and *Zostera* have global distributions in tropical and temperate climates, and *Amphibolis* is a temperate Australian endemic with a similar life history and meadow characteristics to other persistent genera that are more widely distributed (eg, *Posidonia*). Some of the sites have meadows that are enduring, they persist over time, whereas others are transitory, fluctuating between presence and absence of seagrass[23].

With these seagrass meadows in mind, we developed and validated a whole-of-system DBN model[24]. This model captured the conditional probabilistic relationships among population variables (eg, biomass, shoot density), factors relating to resistance (eg, growth and physiology), recovery (eg, physiology, seed and vegetative growth), site conditions (eg, genera present and location characteristics such as depth, climate and tidal regime) and environmental factors (eg, light and water quality). Given site-specific and dredge-specific scenarios and the conditional relationships described above, the posterior-state distributions were computed cumulatively at monthly intervals. These distributions directly encapsulate variations in risk over time, which underpins the resilience criteria adopted here of resistance, recovery and persistence[8]. The modelled response of seagrass to dredging light deprivation stressors revealed opportunities to maximise seagrass resilience through appropriate timing of dredging.

## Results

**Setup and interpretation of modelling study.** Our model enables assessment using site-specific biological and environmental conditions, and comparison between sites. We use

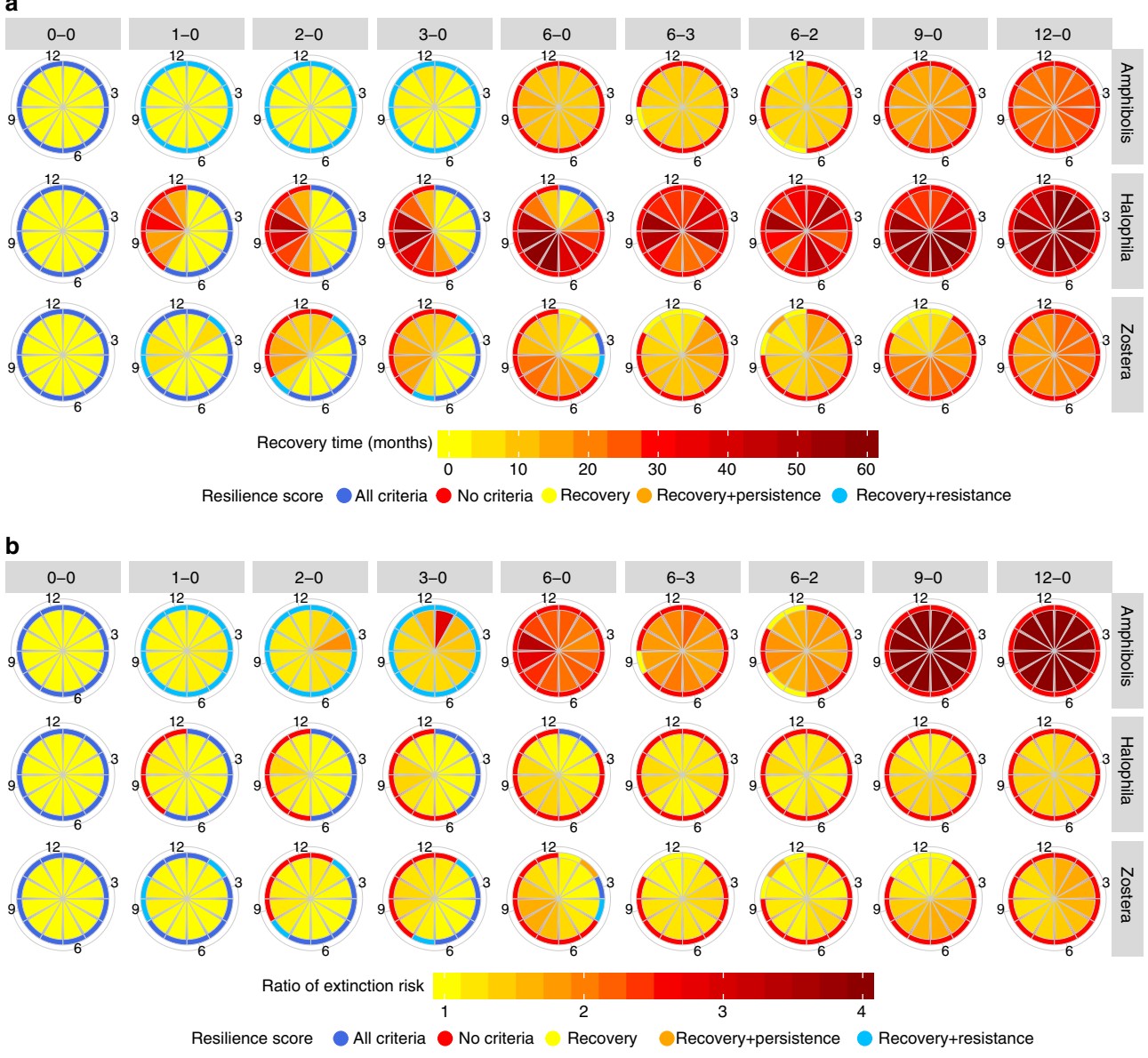

**Fig. 1** Responses of seagrass populations at three sites for eight dredging and one control scenario. The sites (rows) are: persistent *Amphibolis* in Jurien Bay, Australia (top row), colonising *Halophila* at Hay Point, Australia (middle row) and opportunistic *Zostera* in Puget Sound, USA (bottom row). Dredging scenarios (columns), all result in the absence of saturating light. Each pie slice reflects the time to recovery (**a**) and the ratio of the extinction risk compared to baseline (**b**) for dredging starting in that month. Months are indicated by numerals in the outer ring, where 12 denotes December. All results are aligned to Austral summers to enable seasonal comparisons. The outer coloured edge of each pie reports a resilience criteria score (see legend)

scenarios of light reduction and its probability as the primary and proximal stress to seagrass from dredging. Periods and magnitudes of light exposure are directly measurable[22,25] and reflect the combined spatio-temporal effects of dredging, weather and local hydrodynamics. We estimate the probability of above saturation light to capture temporal variations in light for a given site using the number of days of above saturation light in a month ('Methods'). The dredging scenarios modelled vary from 1 to 12 months duration, starting in each month of the year. In the first instance, we assume the complete absence of saturating light during dredging, then compare this to scenarios with 25, 50 and 75% probability of above saturation light during dredging. Also considered are scenarios where dredging is punctuated by non-dredging periods, which could simulate the movement of the dredge away from the area, and which may help improve recovery and reduce extinction risk ('Methods').

Through the application of a whole-of-systems DBN model to scenarios at 28 sites globally, we synthesised state probability and weighted mean responses, and interpreted these in terms of resilience. We developed three criteria based on resistance and recovery[8,9], assessed against contemporary baseline patterns, which may be constant (limit point) or periodically varying (limit cycle)[8,9]. The resistance (minimal loss) criterion was satisfied when there was <20% change in the weighted mean response relative to the baseline immediately after a stress. The recovery criterion was satisfied when the weighted mean recovered to within 20% of the baseline weighted mean within 6 months after the stress had been removed. Finally, the persistence criterion was satisfied when there was no additional increase in the risk of local extinction (probability of the zero state) following the stressor, defined as a ratio of <1.025 between the zero state probability of the response and the baseline.

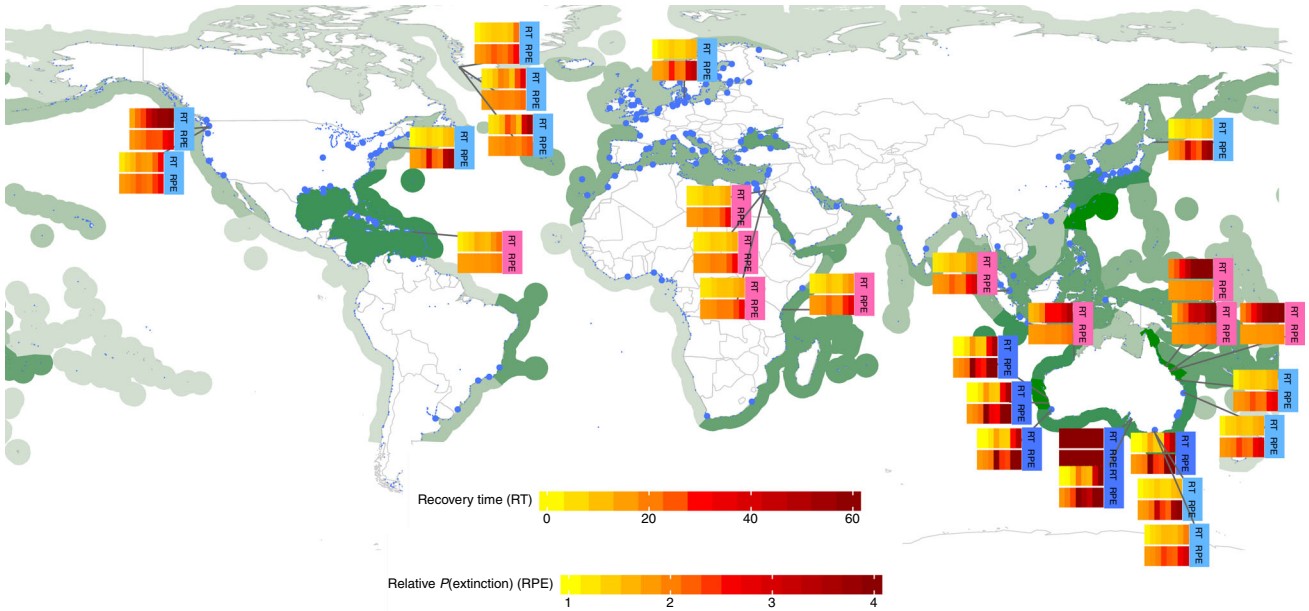

**Fig. 2** Global distribution of seagrasses and ports showing resilience to dredging. Seagrass density[28] is shaded green and ports[29] are indicated by blue dots, where larger dots indicate larger harbours (generated using R software[30] and maps package). For 28 seagrass sites in the vicinity of ports and dredging, one-dimensional heat maps are shown of average recovery time (RT; top panels) and average relative probability of extinction (RPE; bottom panels). Each heat map has eight vertical bars, corresponding to dredging periods denoted as <dredge duration>−<alternating dredge/rest duration> of 1–0, 2–0, 3–0, 6–0, 6–3, 6–2, 9–0 and 12–0 months. Heat map labels are coloured by genus—pink for colonising *Halophila* light blue for opportunistic *Zostera*, and dark blue for persistent *Amphibolis*

For annual meadows, the baseline population could be zero; hence, resilience criteria need to be assessed in reference to a baseline. Twenty per cent was selected as a conservative criterion as it has been used in the management of *Posidonia* meadows[26]. However, because of the flexibility of the modelling approach used here, other thresholds for these criteria could be chosen depending on meadow characteristics and management goals such as where some loss is acceptable. Given these considerations, we used a hierarchical scoring system based on combinations of criteria that were satisfied, from four when all criteria were satisfied, to zero when no criteria were satisfied.

We then used these scores to identify scenarios in which impacts of dredging were minimal, satisfying resistance and recovery (scores 3, 4), or just satisfying recovery (score 1 or 2). Seagrasses are clonal organisms dependent on vegetative growth; hence, maintaining the standing crop is important for their resilience[27] and for maintaining habitat function; thus, resistance and persistence are also important for their resilience. The scores were then used to estimate ecological windows based on when the stress began and its duration. Although the criteria thresholds above were chosen to be conservative, the resulting criteria scores for the 28 global sites reflected the expected resistance or recovery responses for the different life histories[23].

**Ecological windows of seagrass meadows subjected to dredging.** Longer dredging campaigns increased recovery times and the risk of extinction at all sites globally (Fig. 1a, b, respectively, for three representative sites, and Supplementary Figs. 1 and 2). However, more than 3 months of continuous dredging emerged as a key threshold beyond which resilience of seagrasses declined considerably, likely because of their requirement for light and limited ability to store energy for extended periods. Persistent meadows had a marked increase in extinction risk between 3 and 6 months duration, whereas colonising and opportunistic meadows had a major increase in recovery time (Fig. 1). The latter

two types of meadows also displayed distinct windows for dredging commencement, unlike the persistent meadows. When viewed globally (Fig. 2), similarities in the average recovery and extinction risk response also emerged among sites containing genera with common life histories. In general, persistent meadows (eg, meadows near Perth, Australia) exhibited less average resilience (longer recovery time, greater extinction risk) compared to opportunistic meadows (eg, meadows in U.S. or Europe), and these latter opportunistic meadows were less resilient at high latitude sites (eg, Greenland) compared to sites closer to the equator (eg, Brisbane, Australia). Enduring-colonising meadows showed similar average resilience to opportunistic meadows, but transitory colonising meadows, which were all located in Australia in this study, were significantly less resilient on average.

Globally consistent timing of ecological windows across dredging scenarios of similar durations were observed, despite variability in local conditions including depth, subtidal vs. intertidal, baseline light conditions and tropical vs. temperate climate, as well as differences among life histories of the seagrass genera. Therefore, scheduling of dredging according to time of year, where many sites with different local environmental conditions share the same window, provides a powerful tool for maximising resilience under uncertainty. For example, Austral April–May and Boreal October–November is one such seasonally consistent window for opportunistic meadows being dredged for 3 months (Fig. 3) despite meadow locations ranging from 38° south to 64° north latitude and irrespective of differences in light conditions among locations (Fig. 4). For dredging durations of up to 3 months, windows also tended to align with autumn and winter for enduring-colonising and opportunistic meadows. Persistent meadows were generally resistant to small stressors independent of when they occurred while transitory colonising meadows tended to display narrow windows around late summer early autumn (Austral January–February; Boreal July–August for 3 months dredging).

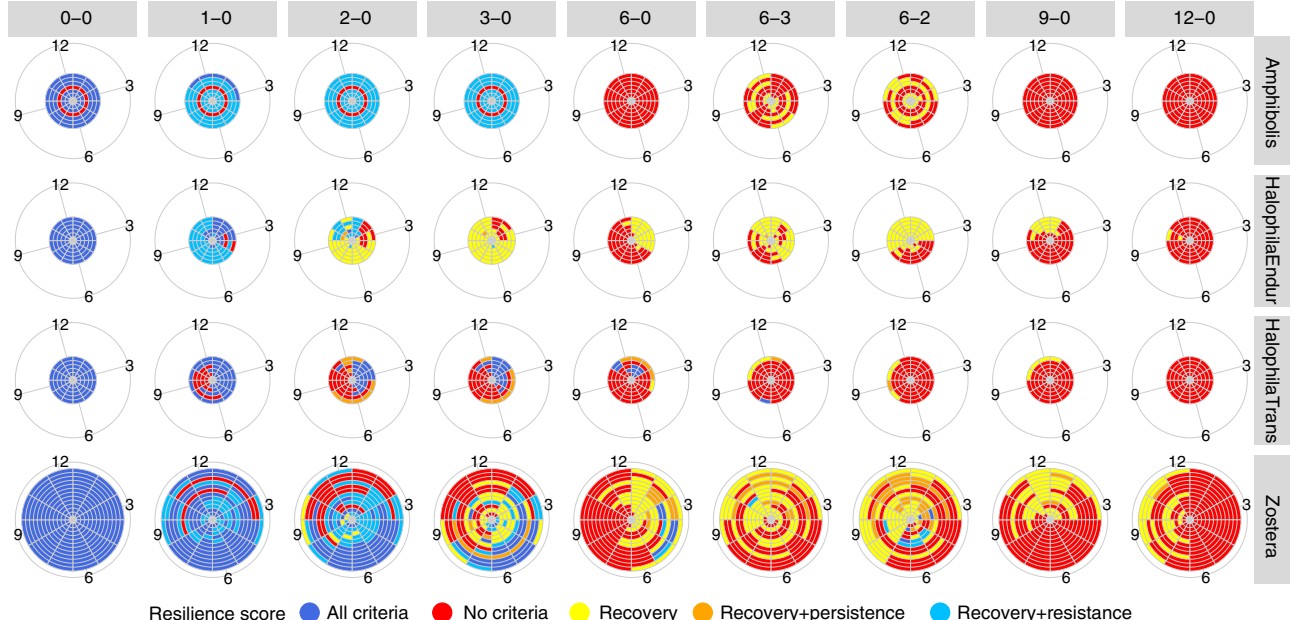

**Fig. 3** Ecological windows for seagrasses by life history strategy/genus, and dredging design. Each ring on a pie corresponds to a site, ordered from southernmost to northernmost going from innermost ring to outermost ring. Resistance, recovery and persistence criteria were considered and criteria scores are colour coded. A score that is not red is considered an ecological window. For individual site results, refer to Supplementary Figs. 1 and 2

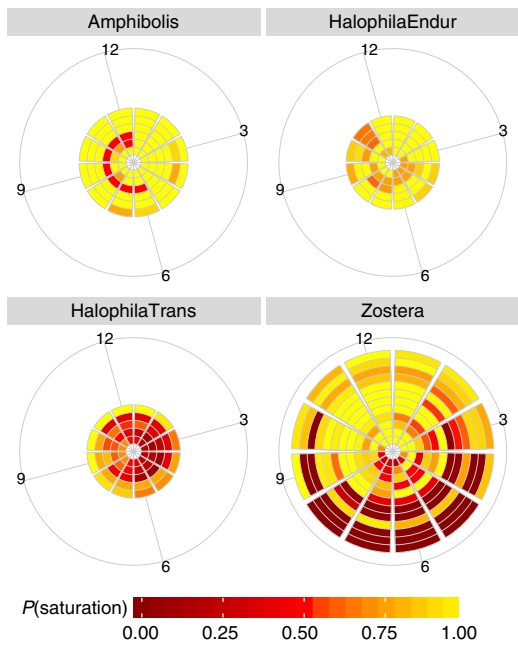

**Fig. 4** The probability of above saturation light conditions for the 28 global sites studied here, grouped by genus and life history of seagrass genera. Each ring represents one site, each pie slice represents one month of the year (labelled numerically in outer ring, 12 corresponds to December), aligned to Austral summers

Using the explanatory nature of DBNs, the ecological reasoning behind these stressor impacts were confirmed by examining model outputs for factors such as shoot density, physiological status, seed density and lateral growth for the three life histories (Supplementary Figs. 3–5). Impact was assessed using site-specific historical baselines comprising biological and environmental patterns, their probability of occurrence and timing ('Methods'). Persistent genera resisted dredging up to 3 months in line with their longer lifespans and high physiological resistance (small drop in population after dredging); beyond 3 months, growth was too slow and seed production too low to recover (Supplementary Fig. 3)[23]. On the other hand, colonising genera invest heavily in seed production (high seed density), rapid vegetative growth and turnover (high variability in shoot density, Supplementary Fig. 4), resulting in shorter recovery times[31] (Fig. 1a). Nonetheless, they have low resistance as shown by a complete transition to yellow scores once dredging duration exceeds 2 months for enduring sites. Yellow indicates an inability to resist but an ability to recover quickly (Fig. 3).

**Variations in windows with stressor intensity.** The preceding analysis assumes the complete absence of light during dredging. However, many practical dredging scenarios may not produce such extreme light stress due to factors including spatial, temporal, mechanical and hydrodynamic effects such as flushing. All these could potentially be forecasted with other models and fed into the DBN we present here. Alternatively, until these more complex models are built and proven to provide greater utility, dredging can be managed using the present model with windows determined by maximum allowable light reduction or equivalently minimum light during dredging. The level of light could be chosen such that the resilience score is at least 1, ie, at minimum the window satisfies the recovery criterion (Fig. 5).

Generally, 25–75% probability of saturating light during dredging was required to achieve recovery for dredging durations of >3 months. Almost all sites could satisfy resilience criteria with 75% light during dredging (the scenario with least light reduction). However, many persistent and enduring-colonising meadows demonstrated year-round resilience up to 6 months at 50% light and even 25% light for enduring-colonising meadows over Spring–Summer with intervening rest periods of 2 months. The benefits of intervening 2 month rest periods also apply to opportunistic meadows which showed longer windows for 25–0% light. Both *Amphibolis* and enduring *Halophila* meadows demonstrated windows for 50% light at 9 months or more duration. On the other hand, transitory *Halophila* did not have

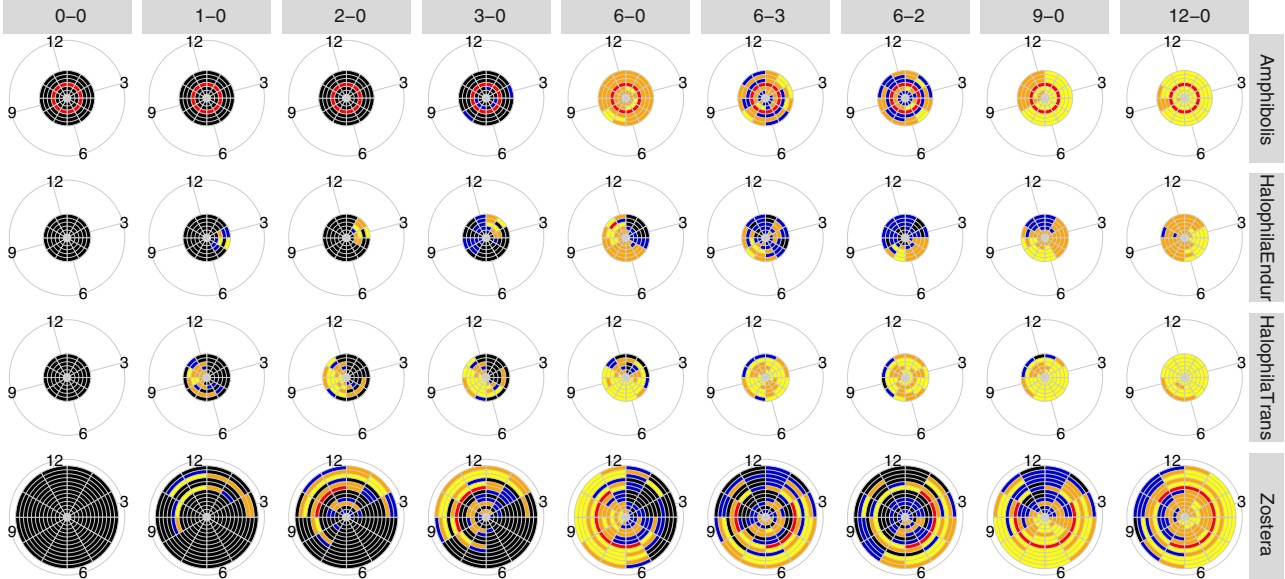

**Fig. 5** Maximum allowable light reduction or, equivalently, minimum light during dredging. Data are grouped by seagrass genus, life history and dredging design. Each ring on a pie corresponds to a site, ordered from southernmost to northernmost from innermost ring to outermost ring. Satisfaction of, at minimum, the recovery criterion and additionally resistance and persistence were considered in determining light reduction thresholds. Minimum light levels are colour coded: black for 0% light during dredging, blue for 25%, orange for 50%, yellow for 75% and red for no light reduction allowed. A score that is not red corresponds to an ecological window. For resilience criteria scores of ecological windows corresponding to 25, 50 and 75% minimum light during dredging, refer to Supplementary Figs. 12–14, respectively

consistent windows for 25 or 50% light, likely due to cumulative effects of poor baseline light conditions (Fig. 4); the only exception was dredging beginning November through January for 6 months duration at 50% light. Opportunistic meadows demonstrated similar windows but could resist slightly greater light deprivation with instances where 25 and 0% light were tolerated at 9 and 12 months duration. In contrast, recovery from seed enabled windows for enduring-colonising meadows with 50% light at up to 12 months.

## Discussion

The results of our model indicate that ecological windows have the potential to help maximise resilience under a range of dredging scenarios. Given that most dredging campaigns are shorter than 6 months[32], we focus on the benefits of ecological windows for these scenarios (Supplementary Table 1). By scheduling stressors, in this case the timing of dredging, an average threefold increase in recovery rate for enduring colonising and opportunistic meadows, and a fourfold increase for transitory colonising meadows was possible. Similarly, the risk of extinction in these same situations was reduced by 17, 35 and 13% respectively. For persistent meadows, there was an average increase of 33% in recovery rate and 21% reduction in risk of extinction. This also highlights the importance of avoiding the application of stress at critical times, which vary locally, to avoid substantial or total loss. Note though that loss of resilience is still likely for capital dredging programs of 6 months or more when there is an absence of light during dredging; unless tight management controls are applied to reduce the magnitude of the stresses imposed such as controlling the allowable reduction in light[33]. Given the global co-distribution of seagrass ecosystems and ports (Fig. 2), potentially many more currently understudied seagrass sites could benefit from ecological windows-based management.

Overall, opportunistic genera were the most resilient; they combine elements of resistance similar to persistent genera, such as a smaller drop in physiological status, with rapid recovery from

fast vegetative growth and recruitment from seeds (Supplementary Fig. 5)[23]. They demonstrated windows of potential recovery denoted by yellow or better resilience scores for 6 or more months of dredging of highest light stress over a significant number of sites (Fig. 3). Similarly, colonising genera also showed recovery potential for up to 6 months dredging but at fewer sites relatively speaking or with narrower windows. In summary, for dredging durations of up to 3 months, enduring colonising and opportunistic meadows demonstrated resilience via recovery criteria, whereas persistent and transitory colonising meadows demonstrated resilience via greater resistance to loss. Unlike persistent meadows, the resistance shown by transitory colonising meadows occurred when dredging aligned with periods when adult seagrasses were absent[34].

These ecological windows were not just a function of the genera examined, nor their life histories[23] including their reproductive periods[14], but were also a function of the timing of growth. Light is a key driver for autotrophs and affects their physiology, growth rates and shoot densities[23]. Sites with long durations and high probabilities of above saturation light (Fig. 4) tended to have longer ecological windows and more resilience to larger stresses (Fig. 3; Supplementary Fig. 4 vs. 6); this was true for many persistent, enduring colonising and opportunistic meadows. In contrast, transitory colonising meadows displayed shorter ecological windows, existed under poorer light conditions, and correspondingly, had shorter growing seasons.

For example, local wet season weather can reduce the probability of saturating light (typical of colonising transitory meadows, Fig. 4). Similarly, ice and poor light in winter can also affect baseline populations. This situation was the case for Greenland sites (outermost three rings for opportunistic sites in Figs. 3 and 4) that displayed comparatively shorter ecological windows and less resilience to larger stresses. Conversely, reduced stress from higher light saturation probabilities during dredging can potentially lengthen ecological windows. In addition, where recovery was possible such as through surviving seeds or population, dredging that concluded at the beginning of the growth season

produced faster recovery than that which concluded at the beginning of the senescent season. In the extreme case, 12 month dredging for *Zostera* in Greenland starting in September (outermost ring, Fig. 5) could recover within 6 months but 6 months dredging could not due to the senescent season (Supplementary Fig. 15). Furthermore, one site had virtually all red scores indicating a complete inability to achieve any resilience criteria (Fig. 3). This site had already been impacted, resulting in a meadow in very poor condition with resulting low resilience and high risk of total loss[31] as correctly predicted by the model. An inability to recover, in the case of this site in particular, corresponded to a shift in the ecosystem baseline[35,36] where local dynamics such as weather patterns and re-suspension of sediment coupled with biological characteristics limited growth and recovery potential.

Although there were consistent windows across sites (Fig. 3), there was significant variation between individual sites especially when considering windows related to minimum light levels during dredging (Fig. 5). The exceptions were 9 and 12 month durations for colonising meadows as they generally tolerated 50 to 75% light during dredging, and persistent meadows as they demonstrate resilience through resistance. Management of light levels during dredging rather than windows will be particularly important for maintenance dredging plumes as these often occur in small areas for short durations close to operations[37], whereas more substantial and longer lived plumes can arise from capital dredging.

Windows-based management has also been applied or could be applied to other ecosystems such as coral reefs. In Western Australia, the current practise requires a 12 day halt to dredging operations (5 days prior and 7 days following) around predicted coral spawning events as a precautionary approach to prevent turbidity from plumes affecting coral reproduction[13]. However, a recent review found over 30 pathways that link turbidity to negative impacts on early life stages of coral beginning with environmental cues that trigger spawning through to settlement and early survival of recruits, indicating that this window may not be wide enough[20]. Our approach of ecological windows could incorporate this information on acute effects on coral reproduction with more chronic effects of turbidity on corals such as light availability affecting photosynthesis, suspended solids affecting feeding and cleaning processes and smothering of corals by sediment[38] to broaden the assessment of impacts to provide a more holistic ecological view.

Our results demonstrate marked variability in realised resilience with stressor timing over monthly time scales, over spatial locations globally, and over stressor durations. This variability highlights the dynamic nature of resilience and the critical need for scenario-based resilience assessment for management. For seagrasses, several studies have highlighted differences in resilience for different seagrass genera[27,39] and differences in potential recovery trajectories for a range of disturbances including climate change[40,41]. These differences are associated with different life history traits[23], but our results also show that they can arise from ecosystem interactions and feedback loops[42], which may produce alternate stable states and impact recovery and resilience[43].

Given the number of factors and complexity of their interactions, the breadth and volume of data needed to support decision makers of many ecosystems including corals, mangroves and seagrasses is not generally available. Our DBN provides a pragmatic approach for capturing whole-of-systems dynamics using expert knowledge and a more feasible, smaller range of observed data. We showed that the model predicted the resultant population probability trajectory with a mean-squared error on the order of 0.02 to 0.05 for data collected from an actual dredging campaign, long-term observational studies and light-shading

experiments ('Methods'). Sensitivity analysis revealed that the most influential variables on population change over time ('Methods') were net change, realised and baseline population, growth, ability to resist and recover, and recruitment from seeds (Supplementary Table 18). This supports the view that the existing population and its growth are dominant factors in maintaining a seagrass meadow[27]. However, seed-based recovery also had significant influence[27], especially on the low state, which suggests it is an important factor for meadows with highly variable populations such as transitory meadows. The model also highlighted that the ability to resist was of crucial importance for persistent species, significantly affecting the likelihood of zero shoot density.

Although existing studies show that resilience is dynamic and can be eroded or restored over time[37,43], changes in realised resilience on monthly or shorter time scales arising from interactions between stressors and ecosystem-baseline dynamics have been poorly understood[14]. Our whole-of-system DBN provides a way to explore these scenarios of stressors, ecosystem dynamics and impact in a probabilistic risk framework[44]. This framework can be applied to a wide range of domains in ecology, but also more generally to the analysis of complex systems such as the movement of passengers through airports[45] and cheetah conservation[46]. Although we have adopted Holling's seminal definition of resilience[8,9], other definitions of resilience could also be adopted in a similar context. For instance, our generative and modular modelling approach could be adapted and/or integrated with other models for the larger scale analysis of nested adaptive cycles (ie, panarchy[47]) over longer time scales.

Our findings about dynamic resilience also support the case for dynamic ocean management (DOM), that changes rapidly in space and time in response to the ecosystem and its users[48,49]. This DBN approach also enables decision makers to explore trade-offs between anthropogenic costs and risks to resilience in real time with a predictive dynamic model that integrates disparate sources of data[48]. However, while both ecological windows and DOM share the same dynamical systems underpinnings, windows are often ecosystem specific and periodic (eg, seasonal[14]), whereas DOM relies on the current state while simultaneously adopting a socio-ecological perspective. In addition, our predictive approach could be applied to the analysis of 'windows of opportunity'[15,16] and especially those that arise due to a pulse disturbance like dredging. Although windows of opportunity are often defined in a socio-ecological context, an ecological window corresponding to a scheduled disturbance could be conceptualised as a type of traction opportunity[16] (eg, a window that allows for recovery). However, here, the opportunity is to better understand resilience rather than the probability of breaking out of an undesirable stable state[50].

There are several additional opportunities for extending our work here. The quantification and interpretation of resilience is still a challenge especially for time to recovery when there are multiple loss and recovery pathways. Similarly, connectivity is currently a poorly understood aspect of seagrass ecosystems that deserves greater attention as new data becomes available[51]. In addition, the integration of our model with forecasting tools such as weather and hydrodynamic models presents an opportunity to test predicted windows for specific sites and form the basis for DOM-based adaptive management[49]. Further integration with other socio-ecological models could support resilience-based management such as 'plan, absorb, recover and adapt' schemes[52] or multi-criteria decision frameworks to make trade-offs against cost, for example[14], or other such approaches[19]. Finally, as more data becomes available, hybrid DBNs that capture both discrete and continuous probability distributions could be explored.

In summary, our analyses provide compelling evidence for the existence and use of ecological windows to help protect and sustainably manage valuable marine resources. In practice, while these models could be customised and integrated with hydrodynamic models for prediction and assessment of local windows, here we have focused on a high level understanding of windows and their potential utility in natural resource management. Although ecological windows were estimated for individual sites, global trends correlated with life histories and environmental conditions such as light still emerged. These windows that are consistent across many locations could be used as a starting point for poorly understood locations scheduled for development. Most importantly, the global applicability of the results presented here, utilising this DBN approach, provides an unprecedented opportunity to help managers evaluate and apply ecological windows based on the predicted resilience of a broad range of biological communities impacted by anthropogenic disturbances.

## Methods

**Study design**. We studied the utility and impact of ecological windows using timing of dredging commencement scenarios and seagrass ecosystem resilience as a canonical example. As resilience[53,54] and hence ecological windows arise from complex interactions between the ecosystem, anthropogenic stressors and timing[8], a multi-faceted, integrative approach is required. First, we developed and validated a whole-of-systems model, the DBN, to enable quantitative, risk-based evaluation of ecological baselines and disturbance scenarios. Second, a series of scenarios were developed to evaluate the impact of timing the start of different dredging campaigns. Finally, we brought modelling and scenarios together to synthesise ecological windows and to quantify their impact in terms of resilience criteria of resistance, recovery and persistence. Despite the existence of many studies of seagrass globally[5,12] or of resilience[17,18], none of these individually capture the sheer scope of data needed to develop ecological windows. Generally, it is infeasible to collect sufficient empirical data to represent the whole spectrum of scenarios over an entire ecosystem[9], thus necessitating the use of expert knowledge and other information.

**Whole-of-systems model**. We used a Bayesian network (BN) and iterative development cycle[46] to structure the elicitation of expert knowledge. BNs, also known as Bayesian belief networks, were suited to our problem because they provide an explicit framework for modelling complex ecosystems with substantial uncertainty. They can combine expert knowledge with data, visualise complex relationships to enhance collaboration, and show good predictive accuracy even with small sample sizes[55,56]. Used increasingly in ecology[42,55,57], BNs encode conditional probabilistic relationships and hence interactions between system factors. They are inherently suited to modelling risk, which is a composition of probability and consequence as part of a scenario[44]. The dynamic component was added to the BN, creating a DBN, to capture cumulative effects and feedback processes characteristic of complex systems[24]. This was extended to a non-homogeneous DBN[58] to model the multiple system transition rules needed to describe feedback. For example, both mortality processes that transition towards extinction and recovery processes that transition away from extinction need to be captured simultaneously[58].

**Model parameterisation and expert knowledge**. The DBN is composed of: (i) the factors identified to be relevant to the management of seagrass under coastal development scenarios, (ii) their relationships in terms of causal influence, (iii) their discretisation and (iv) quantification of conditional probability relationships between them[46,55].

We used expert elicitation, existing policies, guidelines and peer-reviewed literature to identify the factors and processes relevant to managing dredging impacts on seagrass meadows. Expert knowledge is widely applied in ecology especially when there is insufficient empirical data or where such data cannot be gathered until after the fact; often the case for policy and management decisions involving future scenarios[59,60]. Experts are able to identify the key factors relevant to a modelling or management task, and filter, organise and describe data collected through, in our case, decades of experience studying seagrass. We employed a structured, BN-based elicitation and representation approach coupled with validation using expert panels and scenario-based elicitation and evaluation, which has been shown to be highly effective[61]. Empirical validation was also applied to mitigate potential biases and inconsistencies associated with expert knowledge[46].

**Model factors**. The DBN can be found in Supplementary Fig. 7–11 along with node descriptions in Supplementary Table 3; a list of the experts involved in the elicitation of the DBN and network validation are provided in Supplementary Table 5. These comprise a range of internationally recognised seagrass and other

marine and coastal development experts; they are a mix of late-career world's best and mid-career specialists[59]. The elicitation process involves asking the experts about the key variables for management, then asking what factors affect them, and what factors affect those and so on to build up a directed graph of the network[46]. Each factor or node is carefully defined in this process. In summary, the key metrics of interest to management are shoot density (number of shoots or leaf clusters per m$^2$, terminology varies depending on the form of the species studied), and biomass (grams of dry matter per m$^2$)[62,63]. We developed a 34 node network around these two main metrics and light, burial and sediment quality hazards characteristic of dredging. Other hazards not directly related to coastal development such as grazing were not considered in this model but can be easily added due to the modular structure of the network. Similarly, connectivity was not tested in this model although nodes for connectivity (immigrant seeds and vegetative fragments) were added.

**Factor discretisation**. We discretised each factor into states[64], such that continuous variables like biomass are converted into ordinal categories such as high, moderate, low and zero. Categorical factors, such as transitory or persistent meadow types, remain unchanged. Although discretisation is often a requirement of DBN inference[58,64], it also provides a way to capture decision thresholds pertinent to management. These could be thresholds that if exceeded, lead to management actions ranging from increased monitoring to cessation of dredging until levels fall back below the threshold or for a specified period of time[63]. As the output of the DBN is the (discrete, posterior marginal) probability distribution over defined states for each factor, we obtain the risk of meeting or not meeting given decision thresholds. In addition, the somewhat coarse discretisation resolution and associated probability distribution provides a way to capture uncertainty in the system and knowledge of the system; this is a form of epistemic uncertainty[65]. Furthermore, the probabilities used to determine the distribution are continuous so the model can capture a high degree of variability (see 'Sensitivity analysis'). Also present is linguistic uncertainty, which can be mitigated through careful definition of nodes, states and thresholds (Supplementary Table 3).

We discretised all shoot density, biomass and aerial extent nodes, including realised, loss and recovery nodes into high, moderate, low and zero states (Supplementary Table 3). The thresholds for these ordinal categories are defined as a percentage of a reference value. Management of many ecosystems often employ a reference site; for example, the median of a managed site is compared to the 20th and 80th percentile of a reference site(s) under the Australian and New Zealand water quality guidelines for the 'moderate protection' criterion[66]. This enables adaptation and model portability to different meadows with different absolute values in terms of population. Assuming a uniform distribution, we defined the thresholds for high, moderate, low and zero as 81–100, 21–80, 1–20 and 0%. For loss and recovery nodes, the expert elicited discretisation thresholds for high, moderate, low and zero were ≥31, 11–30, 1–10 and 0%, respectively.

In addition, environmental input variables such as light and sediment quality are discretised in terms of their impact on the ecosystem (Supplementary Table 3). Light adequacy for instance is a function of temperature, genera/species, local acclimation and light intensity and duration[67]. Combined with the key role light has in the growth and mortality of seagrass and other benthic habitats, our experts identified probability of above saturation light as the main environmental factor affecting seagrass growth across sites distributed globally. Baseline environmental conditions are defined with the probability of above saturation light, specified for each month of the year. Another advantage of using benthic light at the meadow is that it can be directly measured and encapsulates the overall impact on light due to local hydrodynamic and weather patterns. It thus directly captures baseline conditions prior to dredging; we assume baseline light conditions before and after dredging. Light is also one of the key environmental variables affected by coastal development through dredging and run-off[22,25]. These probabilities for individual sites (Supplementary Table 2) were derived from data via hierarchical linear models[68], expert knowledge or both.

**Elicitation of probabilities**. We adopted both linguistic labels and scenario-based elicitation to maximise cognitive compatibility and to systematically elicit the conditional probabilities used to quantify the DBN from experts[56]. We used linguistic labels of: certainty, extremely likely, very likely, likely, 50/50, unlikely, very unlikely, extremely unlikely and impossible[69]. Given the conditional independence assumptions inherent to a BN[64], scenarios were implemented through comparisons of parent states such as comparing probabilities for fast to slow growth between genera; this helps to ensure consistency. In addition, we employed higher level scenarios relating to plant phenology or growth/senescent seasons (Supplementary Table 19), as well as disturbance scenarios to ensure consistency across the model since it is a whole-of-system model. Nevertheless, elicitation of probabilities is a challenging task due to the number of probabilities needed to parametrise the model and the innate difficulty for human experts to estimate them precisely[56]. Elicitation of probabilities and integration of these with probabilities estimated from data are key areas for future work.

**Model validation**. The model was validated according to each of its constituent components: the factors, the structure, the discretisation and the quantification[46].

| Table 1 Mean-squared error in predicted-state probability for all states and the zero state | | | | | | | |
|---|---|---|---|---|---|---|---|
| | | | | MSE in Predicted State Probability: all states, zero state | | | |
| Site | Genera | Study type | Supplementary table with data | Shoot density | Biomass | Growth | Physiological Status |
| Jurien Bay, Australia[32] | *Amphibolis* | Shading experiment | Light: 6 shoot density, biomass, growth, physiological status: 7 MSE: 8 MSE zero: 9 | 0.02, 0.02 | 0.04, 0.05 | 0.016, 0.03 | 0.05, NA |
| Hay Point, Australia[31] | *Halophila* | Observation of dredging | Light: 2 Biomass: 10 | NA | 0.0011, 0.0029 | NA | NA |
| Salt River Canyon, St Croix[70] | *Halophila* | Long-term monitoring | Light: 2 biomass: 11 | NA | 0.059, 0.026 | NA | NA |
| Gladstone, Australia[40] | *Zostera* | Shading experiment | Light, biomass, growth: 12 MSE: 13 | NA | 0.030, 0.004 | 0.026, 0.020 | NA |
| Puget Sound, USA | *Zostera* | Long-term monitoring | Light: 16 Shoot density: 17 | 0.056, 0.027 | NA | NA | NA |
| Akkeshi Bay, Japan[71] | *Zostera* | Observational study | Shoot density: 14 growth: 15 | 0.012, 0.021 | NA | 0.005, 0.004 | NA |

The first three aspects were validated by a wider expert panel (Supplementary Table 5) and we found that experts had moderate to high confidence in the accuracy and completeness of the model (Supplementary Table 4[68]). Empirical validation was undertaken for each of the three studied genera of *Amphibolis*, *Halophila* and *Zostera*. The probability of above saturation light and site characteristics were used as input. The model predicted-state probabilities for shoot density, biomass, physiological status and lateral growth (depending on data availability) were then compared against observed state probabilities derived from data.

The mean-squared error (MSE) in the predicted-state probabilities compared to observed values was found to be on the order of 0.01 to 0.05, demonstrating a very good to good fit to the data (Table 1). In the case of Jurien Bay *Amphibolis* and Gladstone *Zostera*, the data came from experimental studies of light deprivation effects on seagrass, designed to simulate dredging impacts. These included combinations of different shading durations, shading intensities, and time when shading began according to season or growing vs. senescent periods. Observational studies were also used including monitoring associated with dredging near *Halophila* meadows at Hay Point. Here, the observed data reflected a significant decline in biomass subsequent to dredging, which was picked up by the model. Note that for *Zostera* in Puget Sound, seagrass cover was used to approximate shoot density.

Light saturation probabilities were generated from benthic light measurements using a binomial model[68]. Similarly, measurements of shoot density, biomass, lateral growth and physiological status (using rhizome carbohydrates as a proxy) were converted into state probabilities using hierarchical models. Both models were formulated with the Bayesian framework using Markov Chain Monte Carlo (MCMC)[68]. Although a Receiver Operating Characteristic curve (ROC)[72] or equivalent analysis could be conducted to optimally calibrate data and model predictions, we visually chose a near-optimal calibration that approximates the predicted baseline patterns. Exact solutions were not paramount given the approximate and expert elicited nature of the model. We calculated the MSE across all experiments for the predicted probability of all states compared to data points, and also for just the probability of zero, as avoiding extinction is a key management objective.

Generally, the data had great uncertainty and variability, as expected of naturally occurring complex systems; for a given site, a repeated measure in the same month in a different year can yield significantly different results. In addition, data availability varied greatly across different studies. Also, the available scope of data was small compared to the system as a whole and all the different scenarios and management strategies that could be attempted. This highlights the need for whole-of-systems modelling. We found that the model predominantly demonstrated a good to very good match across the range of life histories, environmental conditions and global locations for population factors and growth. This suggests that the model has correctly captured the core processes underpinning seagrass ecosystems and that the probability of above saturation light serves as an effective proxy of environmental conditions globally.

**Sensitivity analysis.** Sensitivity analysis further confirmed that the model response was sensitive to (ie, non-zero influence) the complex system of factors in the model (Supplementary Table 3). Regression-based sensitivity analysis[73] using boosted regression trees[74] was used to capture non-linear interactions between variables and compute variable influence. Each node and state in the DBN was a time series variable (75 in total) with 96 time slices over 3024 scenarios. The variables were logit transformed DBN posterior probabilities and the tree also included a one-step time lag same as the DBN. Four trees were fitted to the four response variables of high, moderate, low and zero shoot density, and equivalently for biomass. A $R^2$ of 0.99 was achieved for all with mean-squared error of 0.02, 0.02, 0.05 and 0.007 for high to zero states, respectively. We defined the most influential variables arbitrarily as those with weights in the top two orders of magnitude (Supplementary Table 18). Baseline, net change and realised population (at $t-1$) exclusively made up the most influential variables for high and moderate shoot density. Given that the baseline population is calibrated towards moderate to high states ('Methods'), this supports the view that population has a significant role in maintaining population[75]. However, recovery factors including fast growth and high recruitment from seeds feature for low shoot density as they enable transition from zero to low state, supporting recent findings about recovery[27]. By comparison,

ability to resist had a significant effect on the zero state. This further confirms the validity of the model and that discretisation into states does not adversely constrain model response.

**Scenarios.** We applied the validated non-homogeneous DBN to a range of scenarios to synthesise resilience responses and develop ecological windows. We selected a broad range of sites from around the globe, across different latitudes, genera and local conditions; each site must have had light data and known seagrass characteristics (genera, meadow type, habitat, Supplementary Table 3). The purpose of this was to obtain a global picture of ecological windows and ascertain the scale of impact for seagrass and dredging. For each site, the key inputs were the probability of above saturation light, the genera and location specific parameters relating to climate (tropical or temperate), depth and tidal exposure (subtidal or intertidal), and transitory or enduring (persistent) type of meadow[23]. Given the importance of local ecological knowledge[76], we collaborated with experts worldwide. For some sites, such as those in Western Australia, the Red Sea, the US and Greenland, both long-term light data and matching light saturation data were available; thus, the probability of above saturation light was computed directly[68]. This is useful as light saturation thresholds can vary by season[77] and temperature[78]. Some sites, such as Port Phillip Bay, Aininkap and Mombasa, have insufficient light data for statistical modelling. Therefore, we employed expert elicitation based on recorded data to estimate baseline light patterns. In general, light saturation thresholds were obtained from a seminal survey[67]. The model inputs for each site including location parameters, baseline light patterns and local experts engaged are recorded in Supplementary Table 2.

For each site, we generated population trajectories using these baseline conditions for all 34 modelled factors including realised shoot density and biomass. We engaged the experts and reviewed the literature to qualitatively validate the timing of predicted patterns against existing knowledge in terms of: (i) the meadow type (enduring or transitory), (ii) growing season and (iii) flowering and seed production season. For instance, an annual (transitory) meadow should decline to zero population in specific season(s). We found that the literature supports the baseline predictions of the model (Supplementary Table 19). Note that one of the sites in Adelaide suffered compromised light conditions subsequent to dredging, which lead to the loss of that meadow and this was correctly predicted by our model.

Therefore, there is high confidence in model outputs as the model components were individually validated, the model was empirically validated against experiments and dredging campaigns, and expert-checked again after modelling global sites (section 2 and 3).

**Dredging scenarios.** Using the baseline model for each site, we explore ecological windows by developing dredging disturbance scenarios. These were expert elicited and comprised durations of 1, 2, 3, 6, 9 and 12 months with additional consideration for 6 months dredging where dredging alternated with rest periods of 3 months and 2 months. Each dredging design was applied at each month of the year, creating a total of 108 scenarios per site. Dredging is assumed to impact light such that the light saturation probability is reduced to zero during the dredging period (a conservative assumption); outside of the dredging period, light is assumed to follow the baseline pattern. In addition, we consider scenarios where dredging in a given month results in whichever is the smaller out of the baseline probability or either 25, 50 or 75% probability of above saturation light corresponding to a reduction of 75, 50 or 25%, respectively. For each site and scenario, we ran the model and obtained the system response, state probability trajectories over time (eg, Supplementary Figs. 3–6). This response can be sub-divided into three main periods: the initialisation period (to enable settling into the baseline pattern), the stress period (dredging) and the response period. We were specifically interested in the 5 year period after the stress as the Environmental Protection Agency of Western Australia considers a loss permanent if the meadow does not recover within 5 years[62]. For the purposes of assessing deviations from the baseline, we propose a weighted mean approach to aggregate multiple state probability trajectories into a single trajectory for comparison; this weighted mean $\mu(t)$ trajectory over time $t$ is calculated as follows:

$$\mu(t) = p_0(t)\mathbb{I}_{\mu'(t)=0} + (1 - p_0(t))\mu'(t)\mathbb{I}_{\mu'(t)>0},$$

where $p_0(t)$ is the posterior probability of being in a zero state (if such a state exists —eg, zero shoot density), the weighted mean $\mu'(t) = \sum_{j \in \text{states}} p_j(t)\overline{x}_j$, where $p_j(t)$ is the posterior probability of being in state $j$ and $\overline{x}_j$ is the mean or quantile value for state $j$—this value is derived from state thresholds. For example, given thresholds of 81–100% for high shoot density, the quantiles are $\overline{x}_j = \{81, 86, 91, 96, 100\}$ assuming a uniform distribution. Here, we used the median, which equals the mean for a uniform distribution, for recovery and resistance calculations. $\mathbb{I}$ is an indicator function such that: $\mathbb{I} = \begin{cases} 0, \mu'(t) < \alpha \\ 1, \mu'(t) \geq \alpha \end{cases}$, where $\alpha$ is a threshold for being in the zero state.

**Data availability**. The data are available in the Supplementary Information.

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

## Acknowledgements

This research was conducted as part of the Environmental Informatics Programme between QUT and AIMS. This publication was also assisted through an Edith Cowan University Visiting Fellow Grant awarded to Dr Paul Wu. The research was also funded in part by WAMSI as part of the WAMSI Dredging Science Node, Theme 9 (effects of dredging-related pressures on critical ecological processes). We would like to acknowledge the funding support for seagrass and light data from Gladstone Ports Corporation (Gladstone) and North Queensland Bulk Ports Corporation (Hay Point). We would also like to acknowledge the data contributions and local knowledge of: Masahiro Nakaoka, Gidon Winters, David Ball, Dorte Krause-Jensen, Peter Stæhr, Birgit Olesen and Jeffrey Gaeckle; and the expert panel of: Paul Lavery, Paul Erftemeijer, Andy Davis, John Keesing, John Huisman, Dianne McLean, Jessie Short and Ashley Lemmon. The commercial investors and data providers had no role in the data analysis, data interpretation, the decision to publish or in the preparation of the manuscript.

## Author contributions

M.J.C. initiated the study. P.P.-Y.W., Ke.M., M.J.C., Ka.M. and G.A.K. led the writing of the manuscript. All authors contributed to drafts of the manuscript. P.P.-Y.W. developed and validated the statistical models, and analysed and visualised the scenarios. Ke.M. and M.J.C. guided the development and statistical analysis of the models and scenarios. Ka.M., G.A.K., K.C., P.H.Y. and M.A.R. provided data, expert knowledge and ecological analysis of scenarios.

## Additional information

**Competing interests:** The authors declare no competing financial interests. The research was supported in part by investment from Chevron Australia, Woodside Energy Limited, BHP Billiton and WAMSI Joint Venture partners.

