## [Peer Review File · Nature Communications]

Reviewers' comments:

Reviewer #1 (Remarks to the Author):

I find the paper very interesting and important. It provides a rare integration of several methodological and practical concepts that are dear to my heart – dredging, resilience, Bayesian methods. Unfortunately the paper has several logical and methodological weaknesses and also I do not think it has provided a discovery or bridge a knowledge gap that warrant its publication in one of the top general journal like Nature Communication. I believe it is more appropriate for one of the top environmental journal.

My main concern is that the paper presents a naïve view on dredging and its scheduling that makes the model of very limited practical value. It treats dredging as a static and localized process that can take different periods of time to accomplish (up to 12 months). In reality, dredging is done through different methods (mechanical, hopper etc) and is distributed in space and time. It is not clear how integration of localized events is done and how specific dredging methods are integrated in the model. Moreover, dredging windows discussed in the paper are set using local consideration with primary goal of protecting threatened species, authors are wrong stating that dredging windows do not consider site-specificity. Dredging windows are currently set based on risk to the most sensitive species, it is not clear how risk-based scheduling based on local species can be changed to address global issues discussed in the paper.

The model presented in SI shows impacts associated with dredging are realized for the duration of 6 months and longer. Rarely dredging takes so long to implement at one site. In fact, dredging windows may be very short for specific areas. So, it is not clear how conclusion of the study would change if dredging of no more than 6 months is considered (and even 6mo is probably too long).

Framing resilience is another concern that I have. Risk and resilience are subject of intense discussions and authors may not representing it well (see for example IRGC Resource Guide on Resilience <https://www.irgc.org/irgc-resource-guide-on-resilience/> and Linkov et al. Changing the Resilience Paradigm. Nature Climate Change 4:407-409)). What authors define as resistance and persistence is arguably defined as metrics of risk and not resilience. The paper visualizes impact through dashboard of the three criteria that all treated equally. In fact, they probably have very different impact on the overall resilience and color-coded integration is questionable, especially given that only one of the three criteria represents true resilience. The idea of Panarchy for resilience quantification in this case (e.g., Angeler, D.G., et al (2016). Panarchy use in environmental science for risk and resilience planning. Environment, Systems and Decisions 36:225-228.) should be discussed.

The Bayesian Network model is very interesting, but unfortunately it does not consider the most important impact of dredging – suspended sediments concentration. Also, I would assume that pH and oxygen changes during dredging would be important.

Reviewer #2 (Remarks to the Author):

Thank you for giving me the opportunity to review and comment on this work. This paper is excellent and makes a valuable and needed contribution in the efforts to operationalize resilience. Also the contribution highlights a very important and productive ecosystem but poor considered in management strategies, e.g. seagrasses.

The authors make a complex model using a combination of Dynamic Bayesian Networks and scenarios

to investigate resilience variation over time to a specific disturbance. The key elements of resilience considered in the paper are: resistance, recovery and persistence. This is applied to the case of seagrass meadows and dredging with sites represented across the globe.

The results have major management implications. First the authors provide an estimation of resilience in terms of a set of criteria that is met or not, and second this is investigated over time, showing clearly when the disturbance (dredging in this case) becomes severe and when it is possible to handle it or it may be acceptable from a management perspective. This without doubt is welcomed for the benefit of applied management, resilience thinking and understanding ecosystem impacts.

Having said that, one key aspect is however missing from the model, that is water circulation patterns which have a major impact in recovery, in this case from dredging, but plays an equally important role in various impacts in marine environments. This concern should be brought to light by the authors in the discussion section. Since the network has different models (compartments) and sub-models (sub-compartments), additions can be made and a higher level of sophistication developed.

Also I think that an improved discussion about the global validity of the ecological windows found is needed. Having site differences that were not considered in the study (such as water circulation, degree of urbanization, point sources of wastewater, closeness to agricultural sites, etc.) and coarse categorization of the different states, how come that such a consistent pattern is found? Light as determinant for primary producers is well-known and not new as they recognize in the discussion. Two key nodes for the case are the "Ability to resist hazards" and "Accumulated light", I wonder if the Binary States given are too coarse? You treat them as binary states, and for sure that is not happening in reality, could it be a reason to get this rather homogenous global results?

I think that the main text needs a better treatment, specially the resilience criteria should be brought forward here, and how this emergent property is treated by the model and linked to the actual data of recovery and resistance. The discussion section is rather simple and it should be better balanced; less details on the actual case (it is already a lot in results and supplementary information) and more about the wider relevance of this work for seagrass ecosystems and other coastal/marine systems. Some self-criticism here is desirable, given the key factors missing from the model.

Specific comments to the authors:

Review of the manuscript: NCOMMS-16-29157

"Timing anthropogenic stressors to mitigate their impact on marine ecosystem resilience"

SPECIFIC COMMENTS

MAIN TEXT:

Please define coastal development (coastal construction? Increased urbanization in the coastal zones?) Although most of us are aware of what you mean, there are many working in the development community that get upset each time they see this term.

Line 56, add and future paths.

Line 92, explicit about site characteristics

Is there a role also between physiology and recovery? Explain why it is used mainly for resistance

Higher clarity is needed explaining:

1. The three representative sites
2. The global sites
3. The complexity of the method: Network, scenario, resilience criteria and different sources of information (actual data and expert knowledge)

Why a Bayesian Network was used and not a Structural Equation Model?

What are the similarities and differences between ecological windows and windows of opportunities for management intervention?

RESULTS:

Help the reader!

For example, in line 112, and 113 persistent meadows such as.... etc.

In Fig. 1, please remove (methods) and give the criteria, otherwise the paper is too difficult to read.

The resilience criteria are key information needed.

Figure 2. Bottom panels and top panels are wrong given in the text.

Line 124, you have to give more information to the reader about the variability in local conditions, this is part of the key findings that I find so interesting but unclear, why is that?

Fig. 3. Give the resilience criteria again.

METHODS:

Better definition of timing is needed: It can be articulated as when the disturbance starts or the duration of the disturbance, or both??? Please state.

Tables that are referred to as Supplementary Materials are wrong numbered. Table 4 in the text should be Table 3 for example. Please check on all.

You talk about given decision thresholds, specify more. Extend the explanation.

Line 439- we generated responses....to what?

Line 469 – Environmental Protection Agency (USA? Australia?)

The last part of the manuscript e.g. Resilience Criteria and Ecological windows is the most relevant for the understanding of the whole paper. Bring this already in the main text and in the methods move it from the end.

SUPPLEMENTARY INFORMATION

Please check that all numbering is correct.

For the DBN models and figures, it would be nice to give an explanation of the different colors to the reader.

The table of Node definitions is key for the paper as well. Please provide this information in the main text (I mean that this table is key, maybe you should have it in the main text? But probably there is no space for that).

Good luck and thank you for a very interesting reading!

You make a valuable contribution to the advancement of resilience applications and management in general.

Reviewer #3 (Remarks to the Author):

The major claims of the paper include a range of messages such as the value of an “ecological window” management approach, application of Dynamic Bayesian Network (DBN) models based on expert opinion, descriptions of the modeled resilience of seagrasses representative of varying life history strategies, and general implications for dredging of seagrass meadows. The global application of a DBN approach to seagrass modeling is novel. However, as written, the paper seems to strain to place the truly interesting DBN approach to management of dredging into a more theoretical framework. The framework the authors attempt to articulate is the concept of “ecological windows” as a means of managing risk (rather than “environmental windows”) because of the need to consider multiple, interacting stressors.

My reading of the paper did not firmly connect this theoretical framework to the interesting aspects of the modeling and life history findings in a way that was compelling. If the purpose of this paper is to broadly influence thinking regarding this “ecological windows” concept, then there are a few real

challenges. For example, the DBN approach here is applied only to seagrasses and the sole evidence provided in the presented research is the model exercise. Examples of actual applications or case studies are not in evidence; an attempt to broaden the findings to other habitats or generalizable recommendations with application to other organisms for scheduling of stressors around these ecological windows dependent is not attempted. Instead, the concept seems designed to dress up the seagrass modeling efforts. I think this paper would be much more appropriate as a straightforward description of the DBN and its relevance for understanding of resilience of seagrass meadows and management for dredging.

Whether this paper will be of interest to others in the field or influence thinking in the field is a bit confused by the intended audience. If the target is seagrass researchers or those applying DBN approaches, then they could dig through to the specifics of the model and supplementary materials and gain some insights, and the DBN approach is certainly novel. But if a general audience is targeted to generate interest in the "ecological windows" idea as a means for managing anthropogenic disturbance, I feel they may be unsatisfied. In this way, my recommendation above to target the first of these audiences with a longer, more thorough treatment of the model and results (likely in a different journal that permits that format) is more appropriate.

As written, it is sometimes difficult to understand the modeling approach and parameterization to thoroughly evaluate this aspect of the work. Because the Bayesian Network modeling is not common (and truly novel) for the seagrass community, the authors would be more likely to see this approach accepted and used broadly with a more detailed description of the model assumptions, decisions, and organization. For example, little detail is given regarding the methods of collecting information from the expert panel or their work to validate – specifics are important here. To truly garner buy-in from fellow researchers, I also think it will be critical to have a sophisticated discussion of the implications of collapsing continuous variables into ordinal categories. There is a bit of skepticism on my part in those decisions in particular. For the species treated in the model simulations, that are representative of such different strategies and rates of growth, I feel at times that this discretization results in a bit of a "what-you-want-is-what-you-get" model – are we really surprised that the length of the ecological windows differ in the way the simulated results describe for the different life history strategies? At a minimum, some effort to consider a sensitivity analysis that is appropriate for DBNs seems warranted.

The paper is well written, and some of the figures are truly compelling. I applaud the authors in coming up with a creative visual for Figures 1, 3, and 4 to describe their results. However, some of the interesting work in the supplementary materials would benefit from greater attention to the graphical presentations.

Clearly, another way to strengthen the claim that ecological windows are a wise way forward is to take findings from the simulated efforts presented here and test them in case studies, or find analogous examples of success or failure in current management structures that provide ad-hoc evidence that timing is critical. The implementation of specific recommendations from the model is difficult to do in a time frame for this submission. Finding examples of existing management for seagrass or other habitats is likely more feasible.

I was struck by the lack of consideration of the substantial literature regarding resilience of submerged aquatic vegetation. For example, the extensive work of M. Scheffer and colleagues or J Carr's 2012 MEPS paper "Modeling the effects of climate change on eelgrass stability and resilience" and its associated discussion of ecosystem collapse. The substantial literature on restoration and recovery trajectories also seems absent but would be appropriate here (e.g. C. Duarte). I also feel there is great overlap between the "ecological windows" concept and "dynamic ocean management" as defined

by Lewison et al. (2015) and Maxwell et al. (2015).

If the editors feel that a resubmission focused on the model and its findings in terms of seagrass life history strategies is sufficiently novel, then I think this manuscript could be resubmitted to this journal. However, some of the authors participated in an earlier publication (Kilminster et al. 2015) that has some of the general life history findings presented in a different context.

Carr, J.A., D'Odorico, P., McGlathery, K.J. and Wiberg, P.L., 2012. Modeling the effects of climate change on eelgrass stability and resilience: future scenarios and leading indicators of collapse. *Marine Ecology Progress Series*, 448, pp.289-301. And others...2010 stability/bistability paper may be helpful

Duarte, C.M., Conley, D.J., Carstensen, J. and Sánchez-Camacho, M., 2009. Return to Neverland: shifting baselines affect eutrophication restoration targets. *Estuaries and Coasts*, 32(1), pp.29-36.

Lewison, R., Hobday, A.J., Maxwell, S., Hazen, E., Hartog, J.R., Dunn, D.C., Briscoe, D., Fossette, S., O'Keefe, C.E., Barnes, M. and Abecassis, M., 2015. Dynamic ocean management: identifying the critical ingredients of dynamic approaches to ocean resource management. *BioScience*, p.biv018.

Maxwell, S.M., Hazen, E.L., Lewison, R.L., Dunn, D.C., Bailey, H., Bograd, S.J., Briscoe, D.K., Fossette, S., Hobday, A.J., Bennett, M. and Benson, S., 2015. Dynamic ocean management: Defining and conceptualizing real-time management of the ocean. *Marine Policy*, 58, pp.42-50.

Scheffer, M., Carpenter, S., Foley, J.A., Folke, C. and Walker, B., 2001. Catastrophic shifts in ecosystems. *Nature*, 413(6856), pp.591-596. Please note this is just one example, Scheffer has many other papers that would be appropriate to consider here – the 2009 Nature paper, etc.

Unsworth, R.K., Collier, C.J., Waycott, M., McKenzie, L.J. and Cullen-Unsworth, L.C., 2015. A framework for the resilience of seagrass ecosystems. *Marine pollution bulletin*, 100(1), pp.34-46.

van der Heide, T., van Nes, E.H., Geerling, G.W., Smolders, A.J., Bouma, T.J. and van Katwijk, M.M., 2007. Positive feedbacks in seagrass ecosystems: implications for success in conservation and restoration. *Ecosystems*, 10(8), pp.1311-1322.

Reviewers' comments:

Reviewer #1 (Remarks to the Author):

I find the paper very interesting and important. It provides a rare integration of several methodological and practical concepts that are dear to my heart – dredging, resilience, Bayesian methods. Unfortunately the paper has several logical and methodological weaknesses and also I do not think it has provided a discovery or bridge a knowledge gap that warrant its publication in one of the top general journal like Nature Communication. I believe it is more appropriate for one of the top environmental journal.

My main concern is that the paper presents a naïve view on dredging and its scheduling that makes the model of very limited practical value. It treats dredging as a static and localized process that can take different periods of time to accomplish (up to 12 months). In reality, dredging is done through different methods (mechanical, hopper etc) and is distributed in space and time. It is not clear how integration of localized events is done and how specific dredging methods are integrated in the model. Moreover, dredging windows discussed in the paper are set using local consideration with primary goal of protecting threatened species, authors are wrong stating that dredging windows do not consider site-specificity. Dredging windows are currently set based on risk to the most sensitive species, it is not clear how risk-based scheduling based on local species can be changed to address global issues discussed in the paper.

RESPONSE 1:

Thank you for raising this issue, we have conducted significant further study in addressing this. First, we agree that the model itself does not explicitly consider specific dredging mechanics/methods and its distribution over space and time. However, we do capture the dominant stress for seagrass arising from spatio-temporal interactions involving dredging, which is that of light reduction (clarified in Main Text paragraph 3):

“Here we study the impact of scheduled dredging and its associated stressors on the resilience of seagrass meadows as a canonical example. Although dredging produces multiple disturbances including impacts on pH and dissolved oxygen, the predominant stress on seagrass is light reduction arising from suspended sediments{Erftemeijer, 2006 #518}. Periods and levels of light reduction emerge as key variables from complex interactions between spatial and temporal factors including the mechanics of the dredge, relative location of dredging to a seagrass meadow, wind and waves, storms, tides and associated flushing{Erftemeijer, 2006 #518}. As a result, impact assessments of dredging campaigns need to be customised for specific meadows at specific periods in time, and incorporate uncertainty associated with forecasted future conditions.”

Light can be directly measured for a specific dredge campaign and meadow location, and can be used as a direct measure of cumulative historical conditions for establishing environmental baselines (clarified in Methods: Factor Discretisation, paragraph 3):

“Another advantage of using light is that it can be directly measured and encapsulates local hydrodynamic and weather patterns. It thus directly captures baseline conditions prior to dredging; we assume baseline light conditions before and after dredging.”

Our paper aims to assess whether the impact of manipulating the primary driver of seagrass change (light) has an effect on windows. Within this we have looked at different periods of time that the pressure has been applied as well as cyclic application of the pressure with periods of respite to in part allow for some of the potential variations the reviewer suggests. Without demonstrating first that light has a particular effect there would be no purpose in pursuing its cause.

However, we take on board the reviewers comments that dredging is in fact much more complex and often the pressure may be less uniform in time and space due to tides and hydrodynamics as well as variations in dredge operation and plant. We clarify that light is captured as a probability (of being above saturation) for each month and this probability represents variability in conditions during dredging (i.e. not necessarily constant stress) and cumulative spatial and temporal effects. A meadow further away from the dredge might have a lower probability of light reduction for instance. Given our objective to understand ecological windows globally and their customisation locally, we extended our study to compare the ecological windows achieved with our original study (0% light during dredging) and those achieved with 25%, 50% and 75% light during dredging (Main Text, paragraph 7):

“Our model enables assessment using site specific biological and environmental conditions and comparison between sites. We use scenarios of light reduction and its probability as the primary and proximal stress to seagrass from dredging. Periods and magnitudes of light exposure are directly measurable{McMahon, 2011 #628;Erftemeijer, 2006 #518} and reflect the combined spatio-temporal effects of dredging, weather and local hydrodynamics. We estimate the probability of above saturation light to capture temporal variations in light for a given site using the number of days of above saturation light in a month (Methods). The dredging scenarios modelled vary from 1 to 12 months duration, starting in each month of the year. In the first instance, we assume the complete absence of saturating light during dredging, then compare this to scenarios with 25%, 50% and 75% probability of above saturation light during dredging. Also considered are scenarios where dredging is punctuated by non-dredging periods, which could simulate the movement of the dredge away from the area, and which may help improve recovery and reduce extinction risk (Methods).”

In the results paragraph 4-5 and Fig 5, we show how ecological windows can change with light reduction levels and how these levels can be used as a tool for management.

“The preceding analysis assumes the complete absence of light during dredging. However, many practical dredging scenarios may not produce such extreme light stress due to factors including spatial, temporal, mechanical and hydrodynamic effects such as flushing. All these could potentially be forecasted with other models and fed into the DBN we present here. Alternatively, until these more complex model are built and proven to provide greater utility, dredging can be managed using the present model with windows determined by maximum allowable light reduction, or equivalently minimum light during dredging. The level of light could be chosen such that the resilience score is at least 1, i.e. at minimum the window satisfies the recovery criterion (Fig. 5).

Generally, 25% to 75% probability of saturating light during dredging was required to achieve recovery for dredging durations of greater than three months. Almost all sites could satisfy resilience criteria with 75% light during dredging (the scenario with least light reduction). However, many persistent and enduring colonising meadows demonstrated year-round resilience up to six months at 50% light and even 25% light for enduring colonising meadows over Spring-Summer with intervening rest periods of two months. The benefits of intervening two month rest periods also apply to opportunistic meadows which showed longer windows for 25% to 0% light. Both *Amphibolis* and enduring *Halophila* meadows demonstrated windows for 50% light at 9 months or more duration. On the other hand, transitory *Halophila* did not have consistent windows for 25% or 50% light, likely due to cumulative effects of baseline light conditions (Fig. 4); the only exception was dredging beginning November through January for six months duration at 50% light. Opportunistic meadows demonstrated similar windows but could resist slightly greater light deprivation with instances where 25% and 0% light were tolerated at 9 and 12 months duration. In contrast, recovery from seed enabled windows for enduring colonising meadows with 50% light at up to 12 months.”

Fig 5. Maximum allowable light reduction or equivalently minimum light during dredging, for seagrasses by life history strategy/genus, and dredging design. Each ring on a pie corresponds to a site, ordered from southernmost to northernmost from innermost ring to outermost ring. Satisfaction of, at minimum, the recovery criterion and additionally resistance and persistence were considered in determining light reduction thresholds. Minimum light levels are colour coded:

black for 0% light during dredging, blue for 25%, orange for 50%, yellow for 75%, and red for no light reduction allowed. A score that is not red corresponds to an ecological window. For resilience criteria scores of ecological windows corresponding to 25%, 50% and 75% minimum light during dredging, refer to Supplementary Fig. 12 to 14, respectively.

We also clarify the possibility to integrate our model with other models for forecasting (Discussion, second last paragraph):

“In addition, the integration of our model with forecasting tools such as weather and hydrodynamic models presents an opportunity to test predicted windows for specific sites.”

RESPONSE 2:

To the second point about existing environmental windows, we clarify that the challenge for existing work is in understanding the impact of interactions between environmental, stressor and biological systems (Main Text, paragraph 4):

“Our approach is centred around the use of ecological windows: periods during which a specific stressor can occur with minimal impact on resilience. They differ from environmental windows in existing regulatory frameworks (e.g., U.S.A. National Environmental Policy Act 1969) which typically do not consider site-specific biological, environmental and stressor interactions{Suedel, 2008 #638}. Little is currently known about the utility of ecological windows for managing anthropogenic disturbances and resilience{Anthony, 2011 #826;Anthony, 2015 #827}, nor are there currently adequate tools to customise their use for the local condition of ecosystems{Suedel, 2008 #638}. The existing windows framework for management is also restrictive as anthropogenic activities that impact a given ecosystem are prohibited during a time when a critical biological function is thought to occur. Examples include coral spawning in Western Australia (Jones et al 2015) and Pacific herring spawning in San Francisco Bay{Suedel, 2008 #638}. Because windows can emerge from complex interactions between disturbances and ecosystems, data requirements can be onerous, and consequently, have impeded systematic, quantitative estimation of windows{Suedel, 2008 #638}. In the absence of complete data, our Dynamic Bayesian Network (DBN) approach integrates expert knowledge and available data using the established windows framework to estimate the timing and length of windows for specific locations and time periods. We use seagrass meadows as a case study but this approach could be applied widely to other ecosystems.”

In addition, we clarify that the global study is used to: (i) improve our understanding of ecological windows, their trends and applicability, and (ii) use as a starting point for poorly understood

locations scheduled for development (Discussion, last paragraph):

“Although ecological windows were estimated for individual sites, global trends correlated with life histories and environmental conditions such as light still emerged. These windows that are consistent across many locations could be used as a starting point for poorly understood locations scheduled for development.”

The model presented in SI shows impacts associated with dredging are realized for the duration of 6 months and longer. Rarely dredging takes so long to implement at one site. In fact, dredging windows may be very short for specific areas. So, it is not clear how conclusion of the study would change if dredging of no more than 6 months is considered (and even 6mo is probably too long).

RESPONSE 3:

We agree that shorter dredging campaigns are common particularly for maintenance dredging and smaller capital campaigns. The analysis includes examination of dredging over shorter time frames shorter times and in fact, the quoted improvements in recovery time and risk are for 3 months dredging or less (Supplementary Table 1). This has been clarified (Discussion, first paragraph):

“...Given that most dredging campaigns are shorter than 6 months{McCook, 2015 #892}, we focus on the benefits of ecological windows for these scenarios (Supplementary Table 1). By scheduling stressors, in this case the timing of dredging, an average three-fold increase in recovery rate...”

However longer capital dredging projects impacting seagrasses also occur so the longer timeframes examined in this study are of relevance. Examples include:

- Hay Point Australia- 12 months of dredging with chronic light deprivation maintained over seagrasses for at least 8 months (York et al 2015)
- Gladstone Western Basin (Chartrand et al 2016) Capital dredging over more than 2 years (2011-2013)

In fact it is likely that these longer dredging campaigns are where management windows are likely to be of greatest benefit and application. The examples above are for 2 examples in Australia, but these are by no means uncommon throughout Australia and elsewhere around the world.

Framing resilience is another concern that I have. Risk and resilience are subject of intense discussions and authors may not representing it well (see for example IRGC Resource Guide on Resilience <https://www.irgc.org/irgc-resource-guide-on-resilience/> and Linkov et al. Changing the Resilience Paradigm. Nature Climate Change 4:407-409)). What authors define as resistance and persistence is arguably defined as metrics of risk and not resilience. The paper visualizes impact through dashboard of the three criteria that all treated equally. In fact, they probably have very different impact on the overall resilience and color-coded integration is questionable, especially given that only one of the three criteria represents true resilience. The idea of Panarchy for resilience quantification in this case (e.g., Angeler, D.G., et al (2016). Panarchy use in environmental science for risk and resilience planning. Environment, Systems and Decisions 36:225-228.) should be

discussed.

RESPONSE 4:

Thank you for the references. We agree that recovery is an essential part of resilience and we have clarified that the three criteria are not necessarily treated equally - rather, the scoring/colour-coding represents ordinal categories. For example, it only implies that having resistance and recovery is better than just having recovery but not by how much. Furthermore, recovery is present in all non-zero scores. This has been clarified (Main Text, last two paragraphs before Results):

“...Given these considerations, we used a hierarchical scoring system where larger values indicated relatively greater resilience. These scores represent ordinal categories according to the following scheme:

score 4: satisfy resistance, recovery and persistence criteria (dark green)

score 3: satisfy resistance and recovery criteria (green)

score 2: satisfy recovery and persistence criteria (orange)

score 1: satisfy recovery criterion only (yellow)

score 0: no criteria are satisfied (red)

We then used these scores to identify scenarios in which impacts of dredging were minimal, satisfying resistance and recovery (scores 3, 4), or just satisfying recovery (score 1 or 2). Seagrass are clonal organisms dependent on vegetative growth, hence maintaining the standing crop is important for resilience{Kendrick, 2012 #561} (Discussion) and for maintaining habitat function; thus, resistance and persistence are also important for resilience. The scores were then used to estimate ecological windows based on when the stress began and its duration (Fig. 3 and Supplementary Fig. 1 and 2). Although the criteria thresholds above were chosen to be conservative, the resulting criteria scores for the 28 global sites reflected the expected resistance or recovery responses for the different life histories{Kilminster, 2015 #807}.”

For seagrass meadows, which are clonal species reliant in many ways on vegetative growth, the standing crop of the meadow plays a critical role in its ability to recover. This phenomenon is corroborated by our new sensitivity analysis (Discussion, paragraph 8 and Methods: Sensitivity Analysis). In addition, as a habitat for other organisms, maintaining habitat function is also important. Therefore, resistance and persistence are also important criteria for the management for seagrass meadows, in addition to recovery (clarified in extract above).

Finally, we relate our adopted approach for resilience (Halpern, 2007), (Levin, 2008), to panarchy (Main Text, paragraph 2):

“... Although we have adopted a resistance, recovery and persistence approach, our generative model could be adapted to the larger scale analysis of panarchy{Angeler, 2016 #901} such as over longer time scales.”

The Bayesian Network model is very interesting, but unfortunately it does not consider the most important impact of dredging – suspended sediments concentration. Also, I would assume that pH

and oxygen changes during dredging would be important.

RESPONSE 5:

Although suspended sediments are a direct consequence of dredging, the stress for seagrass due to suspended sediments is light (Erftemeijer, 2006). Generally, light disturbance also has the most effect out of all disturbances on seagrass, this has been clarified (Main Text, paragraph 3):

“Although dredging produces multiple disturbances including impacts on pH and dissolved oxygen, the predominant stress on seagrass is light reduction arising from suspended sediments{Erftemeijer, 2006 #518}.”

Reviewer #2 (Remarks to the Author):

Thank you for giving me the opportunity to review and comment on this work. This paper is excellent and makes a valuable and needed contribution in the efforts to operationalize resilience. Also the contribution highlights a very important and productive ecosystem but poor considered in management strategies, e.g. seagrasses.

The authors make a complex model using a combination of Dynamic Bayesian Networks and scenarios to investigate resilience variation over time to a specific disturbance. The key elements of resilience considered in the paper are: resistance, recovery and persistence. This is applied to the case of seagrass meadows and dredging with sites represented across the globe.

The results have major management implications. First the authors provide an estimation of resilience in terms of a set of criteria that is meet or not, and second this is investigated over time, showing clearly when the disturbance (dredging in this case) becomes severe and when it is possible to handle it or it may be acceptable from a management perspective. This without doubt is welcomed for the benefit of applied management, resilience thinking and understanding ecosystem impacts.

Having said that, one key aspect is however missing from the model, that is water circulation patterns which have a major impact in recovery, in this case from dredging, but plays an equally important role in various impacts in marine environments. This concern should be brought to light by the authors in the discussion section. Since the network has different models (compartments) and sub-models (sub-compartments), additions can be made and a higher level of sophistication developed.

RESPONSE 6:

We agree and have first provided clarification about the focus on light as a stressor and its relationship to water circulation/hydrodynamic effects and other factors (Main Text, paragraph 3):

“...Periods and levels of light reduction emerge as key variables from complex interactions between spatial and temporal factors including the mechanics of the dredge, relative location of dredging to a seagrass meadow , wind and waves, storms, tides and associated flushing{Erftemeijer, 2006 #518}...”

We also clarify how light can be directly measured for a specific dredge campaign and meadow location, and can be used as a direct measure of cumulative historical conditions for establishing environmental baselines (clarified in Methods: Factor Discretisation, paragraph 3):

“Another advantage of using light is that it can be directly measured and encapsulates local hydrodynamic and weather patterns. It thus directly captures baseline conditions prior to dredging; we assume baseline light conditions before and after dredging.”

In addition to this, we have extended our study to better understand how different levels of light stress arising from hydrodynamic (e.g. water circulation) and spatio-temporal effects affect windows. Our paper aims to assess whether the impact of manipulating the primary driver of seagrass change (light) has an effect on windows. Without demonstrating first that light has a particular effect there would be no purpose in pursuing its cause. Potentially, light reduction could be used as a management tool to achieve specified resilience requirements. We extended our study to compare the ecological windows achieved with our original study (0% light during dredging) and those achieved with 25%, 50% and 75% light during dredging (Main Text, paragraph 7):

“Our model enables assessment using site specific biological and environmental conditions and comparison between sites. We use scenarios of light reduction and its probability as the primary and proximal stress to seagrass from dredging. Periods and magnitudes of light exposure are directly measurable{McMahon, 2011 #628;Erftemeijer, 2006 #518} and reflect the combined spatio-temporal effects of dredging, weather and local hydrodynamics. We estimate the probability of above saturation light to capture temporal variations in light for a given site using the number of days of above saturation light in a month (Methods). The dredging scenarios modelled vary from 1 to 12 months duration, starting in each month of the year. In the first instance, we assume the complete absence of saturating light during dredging, then compare this to scenarios with 25%, 50% and 75% probability of above saturation light during dredging. Also considered are scenarios where dredging is punctuated by non-dredging periods, which could simulate the movement of the dredge away from the area, and which may help improve recovery and reduce extinction risk (Methods).”

In the results paragraph 4-5 and Fig 5, we show how ecological windows can change with light reduction levels and how these levels can be used as a tool for management.

“The preceding analysis assumes the complete absence of light during dredging. However, many practical dredging scenarios may not produce such extreme light stress due to factors including spatial, temporal, mechanical and hydrodynamic effects such as flushing. All these could potentially be forecasted with other models and fed into the DBN we present here. Alternatively, until these more complex model are built and proven to provide greater utility, dredging can be managed using the present model with windows determined by maximum allowable light reduction, or equivalently minimum light during dredging. The level of light could be chosen such that the resilience score is at least 1, i.e. at minimum the window satisfies the recovery criterion (Fig. 5).

Generally, 25% to 75% probability of saturating light during dredging was required to achieve recovery for dredging durations of greater than three months. Almost all sites could satisfy resilience criteria with 75% light during dredging (the scenario with least light reduction). However, many persistent and enduring colonising meadows demonstrated year-round resilience up to six months at 50% light and even 25% light for enduring colonising meadows over Spring-Summer with intervening rest periods of two months. The benefits of intervening two month rest periods also apply to opportunistic meadows which showed longer windows for 25% to 0% light. Both *Amphibolis* and enduring *Halophila* meadows demonstrated windows for 50% light at 9 months or more duration. On the other hand, transitory *Halophila* did not have consistent windows for 25% or 50% light, likely due to cumulative effects of baseline light conditions (Fig. 4); the only exception was dredging beginning November through January for six months duration at 50% light. Opportunistic meadows demonstrated similar windows but could resist slightly greater light deprivation with instances where 25% and 0% light were tolerated at 9 and 12 months duration. In contrast, recovery from seed enabled windows for enduring colonising meadows with 50% light at up to 12 months.”

Fig 5. Maximum allowable light reduction or equivalently minimum light during dredging, for seagrasses by life history strategy/genus, and dredging design. Each ring on a pie corresponds to a site, ordered from southernmost to northernmost from innermost ring to outermost ring. Satisfaction of, at minimum, the recovery criterion and additionally resistance and persistence were considered in determining light reduction thresholds. Minimum light levels are colour coded:

black for 0% light during dredging, blue for 25%, orange for 50%, yellow for 75%, and red for no light reduction allowed. A score that is not red corresponds to an ecological window. For resilience criteria scores of ecological windows corresponding to 25%, 50% and 75% minimum light during dredging, refer to Supplementary Fig. 12 to 14, respectively.

We also clarify the possibility to integrate our model with other models for forecasting (Discussion, second last paragraph):

“In addition, the integration of our model with forecasting tools such as weather and hydrodynamic models presents an opportunity to test predicted windows for specific sites.”

Also I think that an improved discussion about the global validity of the ecological windows found is needed. Having site differences that were not considered in the study (such as water circulation, degree of urbanization, point sources of wastewater, closeness to agricultural sites, etc.) and coarse categorization of the different states, how come that such a consistent pattern is found? Light as determinant for primary producers is well-known and not new as they recognize in the discussion. Two key nodes for the case are the “Ability to resist hazards” and “Accumulated light”, I wonder if the Binary States given are too coarse? You treat them as binary states, and for sure that is not happening in reality, could it be a reason to get this rather homogenous global results?

RESPONSE 7:

Validity: we have conducted a sensitivity analysis to further support our comprehensive expert and empirical validation of the model (Methods: Model Validation) and discussed this in the main paper (Discussion, paragraph 8 and Methods: Sensitivity Analysis):

Given the number of factors and complexity of their interactions, the breadth and volume of data needed to support decision makers of many ecosystems including corals, mangroves and seagrasses is not generally available. Our DBN modelling provides a pragmatic approach capturing whole-of-systems dynamics using expert knowledge and a more feasible, smaller range of observed data. We showed that the model predicted the resultant population probability trajectory with a mean squared error on the order of 0.02 to 0.05 for data collected from an actual dredging campaign, long term observational studies and light shading experiments (Methods). Sensitivity analysis revealed that the most influential variables on population change over time (Methods) were net change, realised and baseline population, growth, ability to resist and recover, and recruitment from seeds (Supplementary Table 18). This supports the view that existing population and growth are dominant factors in maintaining a seagrass meadow {Kendrick, 2012 #561} with low to high population. However, seed based recovery also had significant influence {Kendrick, 2012 #561} especially on the low state which suggests it is an important factor for meadows with highly variable populations

such as transitory meadows. The model also highlighted that the ability to resist was of crucial importance for persistent species, significantly affecting the likelihood of zero shoot density.

Sensitivity Analysis

Sensitivity analysis further confirmed that the model response was sensitive to (i.e. non-zero influence) the complex system of factors in the model (Supplementary Table 3). Regression based sensitivity analysis{Frey, 2002 #16} using boosted regression trees{Hastie, 2009 #886} was used to capture non-linear interactions between variables and compute variable influence. Each node and state in the DBN was a time series variable (75 in total) with 96 time slices over 3024 scenarios. The variables were logit transformed DBN posterior probabilities and the tree also included a one step time lag same as the DBN. Four trees were fitted to the four response variables of high, moderate, low and zero shoot density and equivalently for biomass. A R^2 of 0.99 was achieved for all with mean squared error of 0.02, 0.02, 0.05 and 0.007 for high to zero states, respectively. We defined the most influential variables arbitrarily as those with weights in the top two orders of magnitude (Supplementary Table 18). Baseline, net change and realised population (at t-1) exclusively made up the most influential variables for high and moderate shoot density. Given that the baseline population is calibrated towards moderate to high states (Methods), this supports the view that population plays a significant role in maintaining population{Paling, 2009 #888}. However, recovery factors including fast growth and high recruitment from seeds feature for low shoot density as they enable transition from zero to low state, supporting recent findings about recovery{Kendrick, 2012 #561}. By comparison, ability to resist had a significant effect on the zero state. This further confirms the validity of the model and that discretisation into states does not adversely constrain model response.

Discretisation: Although nodes such as ability to resist and light have a small number of states, the probabilities used to represent them are continuous. Therefore, the model can capture a high degree of variability (Methods: Factor Discretisation, paragraph 1):

“Furthermore, the probabilities used to determine the distribution are continuous so the model can capture a high degree of variability (see sensitivity analysis).”

Additionally, sensitivity analysis results suggest that the level of discretisation does not adversely affect model dynamics. It reveals that ability to resist does have a significant impact on population trajectories (Response 7: Validity). In addition, light is one of the direct drivers of loss in population, which is revealed to be on the next tier of influence (order of magnitude of relative influence) down from the ones discussed in the main paper (Supplementary Table 18).

However, there are significant differences between sites as supported by the new study about different light reduction levels (Discussion, paragraph 5, and Fig. 5):

Although there were consistent windows across sites (Fig. 3), there was significant

variation between individual sites especially when considering windows related to minimum light levels during dredging (Fig. 5). The exceptions were 9 and 12 month duration for colonising meadows as they generally tolerated 50% to 75% light during dredging, and persistent meadows as they demonstrate resilience through resistance (Results). Management of dredging light disturbance will be particularly important for maintenance dredging plumes as these often occur in small areas close to operations{{Ports Australia}, 2014 #858}{York, 2016 #863} whereas, more substantial plumes can arise from capital dredging.

I think that the main text needs a better treatment, specially the resilience criteria should be brought forward here, and how this emergent property is treated by the model and linked to the actual data of recovery and resistance. The discussion section is rather simple and it should be better balanced; less details on the actual case (it is already a lot in results and supplementary information) and more about the wider relevance of this work for seagrass ecosystems and other coastal/marine systems. Some self-criticism here is desirable, given the key factors missing from the model.

RESPONSE 8:

We outline resilience in paragraph 2 of the main text and have moved Methods: Resilience Criteria and Ecological Windows to the last 3 paragraphs before Results:

“Here, we adopt three widely used criteria for quantifying resilience to an impact: (1) resistance{Levin, 2008 #659}, the loss of individuals and/or species as the result of stress, (2) recovery{Halpern, 2007 #479}, the expected recovery time, and (3) persistence{Levin, 2008 #659}, risk of local extinction (probability of zero population of a species) following stress (Methods).”

“Through the application of a whole-of-systems DBN model to scenarios at 28 sites globally, we synthesised state probability and weighted mean responses and interpreted these in terms of resilience. We developed three criteria based on resistance and recovery{Holling, 1973 #59;Levin, 2008 #659}:

1. Resistance (minimal loss): less than 20% change in the weighted mean response relative to baseline immediately after a stress.
2. Recovery: recovery of the weighted mean to within 20% of the baseline weighted mean within 6 months after the stress has been removed.
3. Persistence: no additional increase in the risk of local extinction (probability of the zero state) following the stressor, defined as a ratio of less than 1.025 between the zero state probability of the response and the baseline.

For annual meadows, the baseline population could be zero, hence multiple criteria need to be used in reference to a baseline. Twenty percent was selected as a conservative criterion as it has been used in the management of *Posidonia* meadows {McMahon, 2011 #831}. However, because of the flexibility of the modelling approach used here, other thresholds for these criteria could be chosen depending on meadow

characteristics and management goals such as where some loss is acceptable. Similarly, both the 6 month recovery threshold and 1.025 risk ratio could be varied for other dredging programs and or situations where some loss is acceptable depending on management goals. Given these considerations, we used a hierarchical scoring system where large values indicated greater resilience. These scores were determined according to the following scheme:

score 4: satisfy criteria 1 and 2 and 3 (dark green)

score 3: satisfy 1 and 2 (green)

score 2: satisfy 2 and 3 (orange)

score 1: satisfy 2 only (yellow)

score 0: no criteria are satisfied (red)

We then used these scores to identify scenarios in which impacts of dredging were minimal, satisfying resistance and recovery (scores 3, 4), or just satisfying recovery (score 1 or 2). The scores were then used to estimate ecological windows based on when the stress began and its duration (Fig. 3 and Supplementary Fig. 1 and 2). Although the criteria thresholds above were chosen to be conservative, the resulting criteria scores for the 28 global sites reflected the expected resistance or recovery responses for the different life histories{Kilminster, 2015 #807}.

As suggested, we have included a wider discussion about windows based management with light thresholds (informed by our new study), application of windows to corals and potentially other ecosystems, and relating our resilience findings to existing literature on the dynamics of seagrass resilience (Discussion, paragraphs 5-7):

Although there were consistent windows across sites (Fig. 3), there was significant variation between individual sites especially when considering windows related to minimum light levels during dredging (Fig. 5). The exceptions were 9 and 12 month duration for colonising meadows as they generally tolerated 50% to 75% light during dredging, and persistent meadows as they demonstrate resilience through resistance (Results). Management of dredging light disturbance will be particularly important for maintenance dredging plumes as these often occur in small areas close to operations{{Ports Australia}, 2014 #858}{York, 2016 #863} whereas, more substantial plumes can arise from capital dredging.

Windows based management has also been applied or could be applied to other ecosystems such as coral reefs. Environmental windows for the management of dredging in the vicinity of coral reefs regularly occurs. In Western Australia the current practise requires a 12 day halt to dredging operations (5 days prior and 7 days following) around predicted coral spawning events as a precautionary approach to prevent turbidity from plumes affecting coral reproduction{Fraser, 2017 #890}. However, a recent review found over 30 pathways that link turbidity to negative impacts on early life stages of coral beginning with environmental cues that trigger spawning through to settlement and early survival of recruits, indicating that this

window may not be wide enough {Jones et al 2015}. Our approach of ecological windows could incorporate this information on acute effects on coral reproduction with more chronic effects of turbidity on corals such as light availability affecting photosynthesis, suspended solids affecting feeding and cleaning processes and smothering of corals by sediment (Jones et al 2016) to broaden the assessment of impacts to provide a more holistic ecological view.

Our results demonstrate marked variability in realised resilience with stressor timing over monthly time scales, over spatial locations globally, and over stressor durations. This variability highlights the dynamic nature of resilience and the critical need for scenario-based resilience assessment for management. For seagrasses several studies have highlighted differences in resilience for different seagrass genera {Kendrick, 2012 #561} {Unsworth, 2015 #893} and differences in potential recovery trajectories for a range of disturbances including climate change {Carr, 2012 #894} {Rasheed, 2011 #856}. These differences are associated with different life history traits {Kilminster, 2015 #807} but also can be due to interacting factors leading to feedback loops {Maxwell, 2015 #782} and alternate stable states that can have a major impact on recovery and resilience {Folke, 2004 #640}.

Finally, we have included some self-review and opportunities for future work (Discussion, second last paragraph):

“There are several opportunities for future extension to our work. The quantification and interpretation of resilience is still a challenge especially for time to recovery when there are multiple loss and recovery pathways. Another issue is connectivity, which is currently a poorly understood aspect of seagrass ecosystems that would be useful to capture as data becomes available. In addition, the integration of our model with forecasting tools such as weather and hydrodynamic models presents an opportunity to test predicted windows for specific sites and form the basis for DOM-based adaptive management {Maxwell, 2015 #896}. Finally, as more data becomes available, hybrid DBNs that capture both discrete and continuous probability distributions could be explored.”

Specific comments to the authors:

Review of the manuscript: NCOMMS-16-29157

"Timing anthropogenic stressors to mitigate their impact on marine ecosystem resilience"

SPECIFIC COMMENTS

MAIN TEXT:

Please define coastal development (coastal construction? Increased urbanization in the coastal zones?) Although most of us are aware of what you mean, there are many working in the development community that get upset each time they see this term.

Line 56, add and future paths.

Line 92, explicit about site characteristics

Is there a role also between physiology and recovery? Explain why it is used mainly for resistance

RESPONSE 9:

We have made the corrections as suggested and clarified coastal development with examples:

“Dredging is a major source of disturbance affecting water quality with hundreds of millions of cubic metres of sediment dredged annually, most of which is associated with coastal development (e.g., port expansion, source of landfill for reclamation projects, coastal construction and shoreline protection and offshore energy exploration {IADC, 2014 #806}).”

We confirm that the physiology node directly affects both growth and seed and is not used more for resistance or recovery. To clarify, it has been coloured as a separate node in Supplementary Figure 7, 8 and 9.

Higher clarity is needed explaining:

1. The three representative sites
2. The global sites
3. The complexity of the method: Network, scenario, resilience criteria and different sources of information (actual data and expert knowledge)

Why a Bayesian Network was used and not a Structural Equation Model?

What are the similarities and differences between ecological windows and windows of opportunities for management intervention?

RESPONSE 10:

Extra discussion and clarification has been provided about the representative sites (Results, paragraph 1):

“As might be expected, longer dredging campaigns increased the risk of extinction and increased recovery times at all sites globally (Fig. 1 for three representative sites, and Supplementary Fig. 1-2). However, more than three months of continuous dredging emerged as a key threshold beyond which resilience of seagrasses declined considerably, likely because of their requirement for light and limited ability to store energy for extended periods. Persistent meadows had a dramatic increase in extinction risk between three and six months duration whilst colonising and opportunistic meadow had a major increase in recovery time (Fig. 1). The latter two types of meadows also displayed distinct windows for dredging commencement, unlike the persistent meadows. ... ”

Similarly for global sites (Results, paragraph 1):

“ When viewed globally (Fig. 2), similarities in the average recovery and extinction risk response also emerged among sites containing genera with common life histories. In general, persistent meadows (e.g. meadows near Perth, Australia) exhibited less average resilience (longer recovery time, greater extinction risk) compared to opportunistic meadows (e.g. meadows in U.S. or Europe), and these latter opportunistic

meadows were less resilient at high latitude sites compared to sites closer to the equator (e.g. Brisbane, Australia). Enduring colonising meadows showed similar average resilience to opportunistic meadows, but transitory colonising meadows, which were all located in Australia in this study, were significantly less resilient on average.”

The overall approach has been clarified with respect to network, scenarios and criteria in the first paragraph of Methods:

“We studied the utility and impact of ecological windows using timing of dredging commencement scenarios and seagrass ecosystem resilience as a canonical example. As resilience{Holling, 2001 #501;Crain, 2008 #821} and hence ecological windows arise from complex interactions between the ecosystem, anthropogenic stressors and timing{Levin, 2008 #659}, a multi-faceted, integrative approach is required. Firstly, we developed and validated a whole-of-systems model, the DBN, to enable quantitative, risk based evaluation of ecological baselines and disturbance scenarios. Secondly, a combination of ecological and coastal development scenarios were developed to evaluate the impact of timing the start of different dredging campaigns. Finally, we brought modelling and scenarios together to synthesise ecological windows and to quantify their impact in terms of resilience criteria of resistance, recovery and persistence. Despite the existence of many studies of seagrass globally{Waycott, 2009 #667;Halpern, 2007 #479} or of resilience{Anthony, 2011 #826;Anthony, 2015 #827}, none of these individually capture the sheer scope of data needed to develop ecological windows{Wu, 2015 #766}. This data must capture a broad range of scenarios of ecosystem dynamics, anthropogenic stressors and timing, in order to identify windows where management requirements for resilience are met. Generally, it is infeasible to collect sufficient empirical data to represent the whole spectrum of scenarios over an entire ecosystem{Holling, 1973 #59}, thus necessitating the use of expert knowledge and other information.”

Use of expert knowledge or data has been explicitly labelled:

Methods: Model parameterisation and knowledge “We used expert elicitation, existing policies, guidelines and peer-reviewed literature to identify the factors and processes relevant to managing dredging impacts on seagrass meadows. ”

Use of expert knowledge to elicit conditional probabilities:

Methods: Elicitation of Probabilities “We adopted both linguistic labels and scenario based elicitation to maximise cognitive compatibility and to systematically elicit the conditional probabilities used to quantify the DBN from experts{Uusitalo, 2007 #594}.”

For light:

Methods: Factor Discretisation last paragraph: “These probabilities for individual sites (Supplementary Table 2) were derived from data via hierarchical linear models{Wu, 2015 #766}, expert knowledge or both. ”

Regarding structural equation modelling, the key difference between it and DBNs is the ability to use expert knowledge, necessary given many gaps in available data. Parameterising a SEM using expert

knowledge is difficult as it requires specification of covariances. A DBN's conditional probabilities on the other hand can be elicited{Pollino, 2007 #512}.

To our understanding, an ecological window differs from a window of opportunity in that the former is proactive, done before dredging commences, whilst the latter is reactive as it is about opportunities that arise during dredging. Thus, the latter includes design of indicators to monitor to trigger interventions.

RESULTS:

Help the reader!

For example, in line 112, and 113 persistent meadows such as.... etc.

In Fig. 1, please remove (methods) and give the criteria, otherwise the paper is too difficult to read.

The resilience criteria are key information needed.

Figure 2. Bottom panels and top panels are wrong given in the text.

Line 124, you have to give more information to the reader about the variability in local conditions, this is part of the key findings that I find so interesting but unclear, why is that?

Fig. 3. Give the resilience criteria again.

RESPONSE 11:

Added example sites to help the reader interpret Fig. 2:

"In general, persistent meadows (e.g. meadows near Perth, Australia) exhibited less average resilience (longer recovery time, greater extinction risk) compared to opportunistic meadows (e.g. meadows in U.S. or Europe), and these latter opportunistic meadows were less resilient at high latitude sites compared to sites closer to the equator (e.g. Brisbane, Australia)."

Labels have been corrected in Fig. 1 and a definition of resilience criteria has been provided for Fig. 1 and 3:

"Resistance, recovery and persistence criteria were considered. Dark green represents all criteria satisfied, green represents resistance and recovery, orange represents recovery and persistence, yellow represents loss but recovery within 6 months, and red represents no criteria satisfied."

Line 124 Local conditions have been clarified (now Results, opening line of paragraph 2):

"Despite variability in local conditions including depth, subtidal versus intertidal, baseline light conditions and tropical versus temperate climate, as well as differences among life history in the seagrass genera, globally consistent timing of ecological windows across dredging scenarios of similar durations were observed. "

METHODS:

Better definition of timing is needed: It can be articulated as when the disturbance starts or the duration of the disturbance, or both??? Please state.

Tables that are referred to as Supplementary Materials are wrong numbered. Table 4 in the text should be Table 3 for example. Please check on all.

You talk about given decision thresholds, specify more. Extend the explanation.

Line 439- we generated responses....to what?

Line 469 – Environmental Protection Agency (USA? Australia?)

The last part of the manuscript e.g. Resilience Criteria and Ecological windows is the most relevant for the understanding of the whole paper. Bring this already in the main text and in the methods move it from the end.

RESPONSE 12:

Timing is about the commencement of dredging which has been clarified (Methods: Study Design):

“We studied the utility and impact of ecological windows using timing of dredging commencement scenarios and seagrass ecosystem resilience as a canonical example. ... Secondly, a combination of ecological and coastal development scenarios were developed to evaluate the impact of timing the start of different dredging campaigns.”

All numbering has been checked.

Decision thresholds were expanded upon:

“These could be thresholds that if exceeded, lead to management actions ranging from increased monitoring to cessation of dredging until levels fall back below the threshold or for a specified period of time{Environmental Protection Authority}, 2005 #622}.”

Clarified line 439:

“For each site, we generated population trajectories using these baseline conditions for all 36 modelled factors including realised shoot density and biomass.”

Line 469:

“as the Environmental Protection Agency of Western Australia”

Methods: Resilience criteria has been moved to the last paragraph before Results:

Through the application of a whole-of-systems DBN model to scenarios at 28 sites globally, we synthesised state probability and weighted mean responses and interpreted these in terms of resilience. We developed three criteria based on resistance and recovery{Holling, 1973 #59;Levin, 2008 #659};

1. Resistance (minimal loss): less than 20% change in the weighted mean response relative to baseline immediately after a stress.
2. Recovery: recovery of the weighted mean to within 20% of the baseline weighted mean within 6 months after the stress has been removed.
3. Persistence: no additional increase in the risk of local extinction (probability of the zero state) following the stressor, defined as a ratio of less than 1.025 between the zero state probability of the response and the baseline.

For annual meadows, the baseline population could be zero, hence multiple criteria need to be used in reference to a baseline. Twenty percent was selected as a conservative criterion as it has been used in the management of *Posidonia* meadows

{McMahon, 2011 #831}. However, because of the flexibility of the modelling approach used here, other thresholds for these criteria could be chosen depending on meadow characteristics and management goals such as where some loss is acceptable. Similarly, both the 6 month recovery threshold and 1.025 risk ratio could be varied for other dredging programs and or situations where some loss is acceptable depending on management goals. Given these considerations, we used a hierarchical scoring system where large values indicated greater resilience. These scores were determined according to the following scheme:

score 4: satisfy criteria 1 and 2 and 3 (dark green)

score 3: satisfy 1 and 2 (green)

score 2: satisfy 2 and 3 (orange)

score 1: satisfy 2 only (yellow)

score 0: no criteria are satisfied (red)

We then used these scores to identify scenarios in which impacts of dredging were minimal, satisfying resistance and recovery (scores 3, 4), or just satisfying recovery (score 1 or 2). The scores were then used to estimate ecological windows based on when the stress began and its duration (Fig. 3 and Supplementary Fig. 1 and 2). Although the criteria thresholds above were chosen to be conservative, the resulting criteria scores for the 28 global sites reflected the expected resistance or recovery responses for the different life histories{Kilminster, 2015 #807}.

SUPPLEMENTARY INFORMATION

Please check that all numbering is correct.

For the DBN models and figures, it would be nice to give an explanation of the different colors to the reader.

The table of Node definitions is key for the paper as well. Please provide this information in the main text (I mean that this table is key, maybe you should have it in the main text? But probably there is no space for that).

RESPONSE 13:

All corrections have been made as suggested, thank you:

Nodes are coloured as follows: white for input nodes, purple for recovery nodes, green for resistance nodes, blue for environmental nodes, yellow for population (shoot density) nodes, and pink for all other nodes.

We are inquiring with the editor about including that table in the main text though it may be better suited in supplementary materials due to length restrictions.

Good luck and thank you for a very interesting reading!

You make a valuable contribution to the advancement of resilience applications and management in general.

Reviewer #3 (Remarks to the Author):

The major claims of the paper include a range of messages such as the value of an “ecological window” management approach, application of Dynamic Bayesian Network (DBN) models based on expert opinion, descriptions of the modeled resilience of seagrasses representative of varying life history strategies, and general implications for dredging of seagrass meadows. The global application of a DBN approach to seagrass modeling is novel. However, as written, the paper seems to strain to place the truly interesting DBN approach to management of dredging into a more theoretical framework. The framework the authors attempt to articulate is the concept of “ecological windows” as a means of managing risk (rather than “environmental windows”) because of the need to consider multiple, interacting stressors.

My reading of the paper did not firmly connect this theoretical framework to the interesting aspects of the modeling and life history findings in a way that was compelling. If the purpose of this paper is to broadly influence thinking regarding this “ecological windows” concept, then there are a few real challenges. For example, the DBN approach here is applied only to seagrasses and the sole evidence provided in the presented research is the model exercise. Examples of actual applications or case studies are not in evidence; an attempt to broaden the findings to other habitats or generalizable recommendations with application to other organisms for scheduling of stressors around these ecological windows dependent is not attempted. Instead, the concept seems designed to dress up the seagrass modeling efforts. I think this paper would be much more appropriate as a straightforward description of the DBN and its relevance for understanding of resilience of seagrass meadows and management for dredging.

RESPONSE 14:

As suggested, firstly, we clarify that we are applying the existing ecological windows framework and provide examples of existing real world applications of the framework for multiple organisms. We clarify that our application is centred on the DBN for seagrass and dredging to learn about both seagrass and the utility of ecological windows (Main Text, paragraph 4 and Discussion, paragraph 6):

“Our approach is centred around the use of ecological windows: periods during which a specific stressor can occur with minimal impact on resilience. They differ from environmental windows in existing regulatory frameworks (e.g., U.S.A. National Environmental Policy Act 1969) which typically do not consider site-specific biological, environmental and stressor interactions{Suedel, 2008 #638}. Little is currently known about the utility of ecological windows for managing anthropogenic disturbances and resilience{Anthony, 2011 #826;Anthony, 2015 #827}, nor are there currently adequate tools to customise their use for the local condition of ecosystems{Suedel, 2008 #638}. The existing windows framework for management is also restrictive as anthropogenic activities that impact a given ecosystem are prohibited during a time when a critical biological function is thought to occur. Examples include coral spawning in Western Australia (Jones et al 2015) and Pacific herring spawning in San Francisco Bay{Suedel, 2008 #638}. Because windows can emerge from complex interactions between disturbances and ecosystems, data requirements can be onerous, and consequently, have impeded systematic, quantitative estimation of windows{Suedel, 2008 #638}. In

the absence of complete data, our Dynamic Bayesian Network (DBN) approach integrates expert knowledge and available data using the established windows framework to estimate the timing and length of windows for specific locations and time periods. We use seagrass meadows as a case study but this approach could be applied widely to other ecosystems.”

“Windows based management has also been applied or could be applied to other ecosystems such as coral reefs. Environmental windows for the management of dredging in the vicinity of coral reefs regularly occurs. In Western Australia the current practise requires a 12 day halt to dredging operations (5 days prior and 7 days following) around predicted coral spawning events as a precautionary approach to prevent turbidity from plumes affecting coral reproduction{Fraser, 2017 #890}. However, a recent review found over 30 pathways that link turbidity to negative impacts on early life stages of coral beginning with environmental cues that trigger spawning through to settlement and early survival of recruits, indicating that this window may not be wide enough {Jones et al 2015}. Our approach of ecological windows could incorporate this information on acute effects on coral reproduction with more chronic effects of turbidity on corals such as light availability affecting photosynthesis, suspended solids affecting feeding and cleaning processes and smothering of corals by sediment (Jones et al 2016) to broaden the assessment of impacts to provide a more holistic ecological view.”

Whether this paper will be of interest to others in the field or influence thinking in the field is a bit confused by the intended audience. If the target is seagrass researchers or those applying DBN approaches, then they could dig through to the specifics of the model and supplementary materials and gain some insights, and the DBN approach is certainly novel. But if a general audience is targeted to generate interest in the “ecological windows” idea as a means for managing anthropogenic disturbance, I feel they may be unsatisfied. In this way, my recommendation above to target the first of these audiences with a longer, more thorough treatment of the model and results (likely in a different journal that permits that format) is more appropriate.

As written, it is sometimes difficult to understand the modeling approach and parameterization to thoroughly evaluate this aspect of the work. Because the Bayesian Network modeling is not common (and truly novel) for the seagrass community, the authors would be more likely to see this approach accepted and used broadly with a more detailed description of the model assumptions, decisions, and organization. For example, little detail is given regarding the methods of collecting information from the expert panel or their work to validate – specifics are important here. To truly garner buy-in from fellow researchers, I also think it will be critical to have a sophisticated discussion of the implications of collapsing continuous variables into ordinal categories. There is a bit of skepticism on my part in those decisions in particular. For the species treated in the model simulations, that are representative of such different strategies and rates of growth, I feel at times that this discretization results in a bit of a “what-you-want-is-what-you-get” model – are we really surprised that the length of the ecological windows differ in the way the simulated results describe for the different life history strategies? At a minimum, some effort to consider a sensitivity analysis that is appropriate for DBNs seems warranted.

RESPONSE 15:

We have attempted to clarify the methodology overall (Method) and provide further details about the elicitation process as suggested. However, further guidance would be appreciated to address any specific concerns. A selection of excerpts are provided below (Method: Model Factors, Supplementary Table 4):

“The elicitation process involves asking the experts about the key variables for management, then asking what factors affect them, and what factors affect those and so on to build up a directed graph of the network{Johnson, 2010 #26}. Each factor or node is carefully defined in this process. ”

Supplementary Table 4. Expert Validation

Count of experts and the level of confidence they expressed about different aspects of the model. Input factors relate to nodes coloured white in Supplementary Fig. 7, recovery factors to Supplementary Fig. 8, resistance factors to Supplementary Fig. 9, and population (shoot density and biomass) coloured yellow in Supplementary Fig. 7 through 9 and 11{Wu, 2015 #766}.

In addition, a summary of model assumptions as discussed in Methods, is provided in Supplementary Text:

Supplementary Text: Modelling Assumptions

Assumes constant connectivity effects

Assumes only stresses are light related (zero sediment quality stress, burial stress, grazing or other stresses)

Assumes probability of above saturation light during dredging period to be the minimum of baseline probability for that month and the light stress scenario of 0%,

25%, 50% or 75%

Assumes negligible effects of competition e.g. from macroalgae

Assumes negligible effects of epiphytes

Assumes adequate nutrition

Sensitivity Analysis: we have conducted a sensitivity analysis to further support our comprehensive expert and empirical validation of the model (Methods: Model Validation) and discussed this in the main paper (Discussion, paragraph 8 and Methods: Sensitivity Analysis):

Given the number of factors and complexity of their interactions, the breadth and volume of data needed to support decision makers of many ecosystems including corals, mangroves and seagrasses is not generally available. Our DBN modelling provides a pragmatic approach capturing whole-of-systems dynamics using expert knowledge and a more feasible, smaller range of observed data. We showed that the model predicted the resultant population probability trajectory with a mean squared error on the order of 0.02 to 0.05 for data collected from an actual dredging campaign, long term observational studies and light shading experiments (Methods). Sensitivity analysis revealed that the most influential variables on population change over time (Methods) were net change, realised and baseline population, growth, ability to resist and recover, and recruitment from seeds (Supplementary Table 18). This supports the view that existing population and growth are dominant factors in maintaining a seagrass meadow (Kendrick, 2012 #561) with low to high population. However, seed based recovery also had significant influence (Kendrick, 2012 #561) especially on the low state which suggests it is an important factor for meadows with highly variable populations such as transitory meadows. The model also highlighted that the ability to resist was of crucial importance for persistent species, significantly affecting the likelihood of zero shoot density.

Sensitivity Analysis

Sensitivity analysis further confirmed that the model response was sensitive to (i.e. non-zero influence) the complex system of factors in the model (Supplementary Table 3). Regression based sensitivity analysis (Frey, 2002 #16) using boosted regression trees (Hastie, 2009 #886) was used to capture non-linear interactions between variables and compute variable influence. Each node and state in the DBN was a time series variable (75 in total) with 96 time slices over 3024 scenarios. The variables were logit transformed DBN posterior probabilities and the tree also included a one step time lag same as the DBN. Four trees were fitted to the four response variables of high, moderate, low and zero shoot density and equivalently for biomass. A R^2 of 0.99 was achieved for all with mean squared error of 0.02, 0.02, 0.05 and 0.007 for high to zero states, respectively. We defined the most influential variables arbitrarily as those with weights in the top two orders of magnitude (Supplementary Table 18). Baseline, net change and realised population (at $t-1$) exclusively made up the most influential variables for high and moderate shoot density. Given that the baseline population is calibrated towards moderate to high states (Methods), this supports the view that population plays a significant role in maintaining population (Paling, 2009 #888).

However, recovery factors including fast growth and high recruitment from seeds feature for low shoot density as they enable transition from zero to low state, supporting recent findings about recovery {Kendrick, 2012 #561}. By comparison, ability to resist had a significant effect on the zero state. This further confirms the validity of the model and that discretisation into states does not adversely constrain model response.

Discretisation: Although nodes such as ability to resist and light have a small number of states, the probabilities used to represent them are continuous. Therefore, the model can capture a high degree of variability (Methods: Factor Discretisation, paragraph 1):

“Furthermore, the probabilities used to determine the distribution are continuous so the model can capture a high degree of variability (see sensitivity analysis).”

Additionally, sensitivity analysis results suggest that the level of discretisation does not adversely affect model dynamics. It reveals for example that ability to resist does have a significant impact on population trajectories (Response 15: Model Validity). In addition, light is one of the direct drivers of loss in population, which is revealed to be on the next tier of influence (order of magnitude of relative influence) down from the ones discussed in the main paper (Supplementary Table 18).

However, there are significant differences between sites as supported by the new study (added in response to reviewer 1) about different light reduction levels (Discussion, paragraph 5, and Fig. 5):

Although there were consistent windows across sites (Fig. 3), there was significant variation between individual sites especially when considering windows related to minimum light levels during dredging (Fig. 5). The exceptions were 9 and 12 month duration for colonising meadows as they generally tolerated 50% to 75% light during dredging, and persistent meadows as they demonstrate resilience through resistance (Results). Management of dredging light disturbance will be particularly important for maintenance dredging plumes as these often occur in small areas close to operations {{Ports Australia}, 2014 #858}{York, 2016 #863} whereas, more substantial plumes can arise from capital dredging.

The paper is well written, and some of the figures are truly compelling. I applaud the authors in coming up with a creative visual for Figures 1, 3, and 4 to describe their results. However, some of the interesting work in the supplementary materials would benefit from greater attention to the graphical presentations.

Clearly, another way to strengthen the claim that ecological windows are a wise way forward is to take findings from the simulated efforts presented here and test them in case studies, or find analogous examples of success or failure in current management structures that provide ad-hoc evidence that timing is critical. The implementation of specific recommendations from the model is difficult to do in a time frame for this submission. Finding examples of existing management for seagrass or other habitats is likely more feasible.

I was struck by the lack of consideration of the substantial literature regarding resilience of submerged aquatic vegetation. For example, the extensive work of M. Scheffer and colleagues or J Carr's 2012 MEPS paper "Modeling the effects of climate change on eelgrass stability and resilience" and its associated discussion of ecosystem collapse. The substantial literature on restoration and recovery trajectories also seems absent but would be appropriate here (e.g. C. Duarte).

I also feel there is great overlap between the "ecological windows" concept and "dynamic ocean management" as defined by Lewison et al. (2015) and Maxwell et al. (2015).

Response 16:

We review and discuss an analogous real world application of windows to the management of dredging on corals. We show that our work for seagrass, which has been bolstered with a new study to incorporate practical application to management (see Response 1 to reviewer 1), could address some of the current challenges of windows based management for corals. In addition, we link our resilience based approach to windows, providing additional discussion and reference to the literature around resilience and recovery as it pertains to seagrasses and submerged aquatic vegetation (Discussion, paragraphs 5-7):

"Although there were consistent windows across sites (Fig. 3), there was significant variation between individual sites especially when considering windows related to minimum light levels during dredging (Fig. 5). The exceptions were 9 and 12 month duration for colonising meadows as they generally tolerated 50% to 75% light during dredging, and persistent meadows as they demonstrate resilience through resistance (Results). Management of dredging light disturbance will be particularly important for maintenance dredging plumes as these often occur in small areas close to operations{{Ports Australia}, 2014 #858}{York, 2016 #863} whereas, more substantial plumes can arise from capital dredging.

Windows based management has also been applied or could be applied to other ecosystems such as coral reefs. Environmental windows for the management of dredging in the vicinity of coral reefs regularly occurs. In Western Australia the current practise requires a 12 day halt to dredging operations (5 days prior and 7 days following) around predicted coral spawning events as a precautionary approach to prevent turbidity from plumes affecting coral reproduction{Fraser, 2017 #890}. However, a recent review found over 30 pathways that link turbidity to negative impacts on early life stages of coral beginning with environmental cues that trigger spawning through to settlement and early survival of recruits, indicating that this window may not be wide enough {Jones et al 2015}. Our approach of ecological windows could incorporate this information on acute effects on coral reproduction with more chronic effects of turbidity on corals such as light availability affecting photosynthesis, suspended solids affecting feeding and cleaning processes and smothering of corals by sediment (Jones et al 2016) to broaden the assessment of impacts to provide a more holistic ecological view.

Our results demonstrate marked variability in realised resilience with stressor timing

over monthly time scales, over spatial locations globally, and over stressor durations. This variability highlights the dynamic nature of resilience and the critical need for scenario-based resilience assessment for management. For seagrasses several studies have highlighted differences in resilience for different seagrass genera {Kendrick, 2012 #561} {Unsworth, 2015 #893} and differences in potential recovery trajectories for a range of disturbances including climate change {Carr, 2012 #894} {Rasheed, 2011 #856}. These differences are associated with different life history traits {Kilminster, 2015 #807} but also can be due to interacting factors leading to feedback loops {Maxwell, 2015 #782} and alternate stable states that can have a major impact on recovery and resilience {Folke, 2004 #640}. ”

Additionally, our results and approach support the application of Dynamic Ocean Management (DOM) and highlight similarities and subtle differences between ecological windows and DOM (Discussion, third last paragraph):

“Although existing studies show that resilience is dynamic and can be eroded or restored over time {Folke, 2004 #640; Gunderson, 2000 #533}, changes in realised resilience on monthly or shorter time scales arising from interactions between stressors and ecosystem baseline dynamics have been poorly understood {Suedel, 2008 #638}. Our whole-of-system DBN provides a way to explore these scenarios of stressors, ecosystem dynamics and impact in a probabilistic risk framework {Kaplan, 1981 #483}. It also supports the case for Dynamic Ocean Management (DOM), management that changes rapidly in space and time in response to the ecosystem and its users {Lewison, 2015 #897} {Maxwell, 2015 #896}. The DBN approach enables decision makers to make trade-offs between anthropogenic costs and risks to resilience in real time with a predictive dynamic model that integrates disparate sources of data {Lewison, 2015 #897}. However, whilst both ecological windows and DOM share the same dynamical systems underpinnings, windows are often ecosystem specific and periodic (e.g. seasonal {Suedel, 2008 #638}), whereas DOM relies on the current state and takes a socio-ecological perspective.”

Finally, we discuss observations of ecosystem trajectories in terms of recovery versus loss, baseline shifts (Discussion, paragraph 4):

“... Furthermore, two sites had virtually all red scores indicating a complete inability to achieve any resilience criteria (Fig. 3). These were sites that had already been impacted by dredging, resulting in a meadow in very poor condition with resulting low resilience and high risk of total loss {York, 2015 #781} as correctly predicted by the model. An inability to recover, in the case of these two sites in particular, corresponded to a shift in the ecosystem baseline {Duarte, 2009 #898} where local dynamics such as weather patterns and re-suspension of sediment coupled with biological characteristics limited growth and recovery potential.”

If the editors feel that a resubmission focused on the model and its findings in terms of seagrass life

history strategies is sufficiently novel, then I think this manuscript could be resubmitted to this journal. However, some of the authors participated in an earlier publication (Kilminster et al. 2015) that has some of the general life history findings presented in a different context.

Carr, J.A., D'Odorico, P., McGlathery, K.J. and Wiberg, P.L., 2012. Modeling the effects of climate change on eelgrass stability and resilience: future scenarios and leading indicators of collapse. *Marine Ecology Progress Series*, 448, pp.289-301. And others...2010 stability/bistability paper may be helpful

Duarte, C.M., Conley, D.J., Carstensen, J. and Sánchez-Camacho, M., 2009. Return to Neverland: shifting baselines affect eutrophication restoration targets. *Estuaries and Coasts*, 32(1), pp.29-36.

Lewison, R., Hobday, A.J., Maxwell, S., Hazen, E., Hartog, J.R., Dunn, D.C., Briscoe, D., Fossette, S., O'Keefe, C.E., Barnes, M. and Abecassis, M., 2015. Dynamic ocean management: identifying the critical ingredients of dynamic approaches to ocean resource management. *BioScience*, p.biv018.

Maxwell, S.M., Hazen, E.L., Lewison, R.L., Dunn, D.C., Bailey, H., Bograd, S.J., Briscoe, D.K., Fossette, S., Hobday, A.J., Bennett, M. and Benson, S., 2015. Dynamic ocean management: Defining and conceptualizing real-time management of the ocean. *Marine Policy*, 58, pp.42-50.

Scheffer, M., Carpenter, S., Foley, J.A., Folke, C. and Walker, B., 2001. Catastrophic shifts in ecosystems. *Nature*, 413(6856), pp.591-596. Please note this is just one example, Scheffer has many other papers that would be appropriate to consider here – the 2009 Nature paper, etc.

Unsworth, R.K., Collier, C.J., Waycott, M., McKenzie, L.J. and Cullen-Unsworth, L.C., 2015. A framework for the resilience of seagrass ecosystems. *Marine pollution bulletin*, 100(1), pp.34-46.

van der Heide, T., van Nes, E.H., Geerling, G.W., Smolders, A.J., Bouma, T.J. and van Katwijk, M.M., 2007. Positive feedbacks in seagrass ecosystems: implications for success in conservation and restoration. *Ecosystems*, 10(8), pp.1311-1322.

Reviewers' comments:

Reviewer #1 (Remarks to the Author):

Even though the response is extensive and authors tried really hard, they are still confused about the use of dredging in real world and in the issue of resilience and risk. I still recommend a major revision, otherwise publishing paper will just add to the confusion in the field and will be quite detrimental to people who really trying to change the way dredging is done in practice.

Authors keep insisting, even in the paper abstract, that resilience is "aka risk, recovery and resistance" which clearly is not. In my last review, I recommend looking at extensive resilience literature summarized in the IRGC guide book, and I specifically recommended papers doi: 10.1038/nclimate2227, doi: 10.1038/srep19540 that were clearly not reviewed by authors.

Moreover, the paper keeps insisting on superiority of their approach without discussing limitation and are using baseless statements like : "Our results contrast with current approaches that focus on a single ecological event without consideration of resilience and interacting factors." Current dredging windows are developed based on extensive consultations with multiple stakeholders integrating ecological modeling and decision analytical tools. Of course the paper presents potentially interesting addition to the relevant science, but it should be placed in appropriate way in the body of current literature and practice. By the way, of course site-specific information is carefully looked at in setting dredging windows. Please review multiple Technical Notes posted at the USACE ERDC web site on the relevant issues.

Reviewer #2 (Remarks to the Author):

Dear authors,

Thank you for the revised version of the paper. I am satisfied with the answers and amendments done to the paper. And I recommend this for publication given that you address some minor further points:

1. The first part of the main text, page 1, has no flow. It is written in telegraphic style and reads badly. This can be easily changed.
2. I think that the links you do to "Panarchy" are still unclear (Lines 68-70)
3. While you explain the difference between environmental and ecological windows, you do not address the issue of windows of opportunities in relation to ecological windows. This must be done. The reason is that this paper is about resilience and in the resilience literature windows of opportunity are broadly used. So the new "ecological windows" have to be situated in this context also.
4. Line 79 please change flushing for water circulation
5. Lines 100-101 repetitive
6. Line 109, about Posidonia is this really global? I don't think so, it's mainly from Australia and the Mediterranean. Please check.
7. Lines 553-554 this is a strong statement without any evidence, what are the references for this? Either you provide them or remove.
8. There are several papers that you can use to address the links between your ecological windows and "windows of opportunities" in the resilience literature. I recommend you to look at:
Nyström et al. 2012. In Ecosystems. This is about marine systems and should be very helpful.
Moon et al. 2014. Conservation Ecology. This can be useful as it addresses the know GBR case in Australia.

Other more general and central in the resilience literature are:

Folke et al. 2010. Ecology and Society.

Olsson P. et al. 2006. Ecology and Society.
Young O. 2010. Global Environmental Change.

Thank you for an excellent and interesting work!

Reviewer #3 (Remarks to the Author):

I am most satisfied in the authors' response to my review to focus the "ecological windows" concept and writing in this manuscript. Providing the coral reef example was illuminating and I believe will be helpful for the reader. The authors also expanded their discussion to couch the approach in Dynamic Ocean Management and to also include a more sophisticated discussion of restoration trajectories.

I appreciate the sensitivity analysis outlined in the new manuscript, it addresses some of my concerns. The responses to Reviewer #2's more specific comments regarding the expert opinion work was also helpful. However, even with the stated assumptions and the sensitivity analysis it is still not transparent to me where the "tender" points are in the model. In most modeling papers those are more clear to me than what is presented here - perhaps because I am more accustomed to looking at frequentist-based statistical models, or system dynamics simulations that use series of ODEs. I suspect, but am not sure, that the assignment of the conditional probabilities is the area with the most uncertainty? Having a statement in the revised manuscript identifying areas that are worthy of next generation investigation and improvement for this approach - or even broadly identifying areas where this modeling approach has its challenges, will be very useful to an audience introduced to these methods via this manuscript.

Reviewers' comments:

Reviewer #1 (Remarks to the Author):

Even though the response is extensive and authors tried really hard, they are still confused about the use of dredging in real world and in the issue of resilience and risk. I still recommend a major revision, otherwise publishing paper will just add to the confusion in the field and will be quite detrimental to people who really trying to change the way dredging is done in practice.

Authors keep insisting, even in the paper abstract, that resilience is "aka risk, recovery and resistance" which clearly is not. In my last review, I recommend looking at extensive resilience literature summarized in the IRGCguide book, and I specifically recommended papers doi: 10.1038/nclimate2227, doi: 10.1038/srep19540 that were clearly not reviewed by authors.

RESPONSE 1:

Thank you for raising this issue. Our focus is on the ecological impact of ecological windows and the application of models to understand and predict them to support management, not specifically how to plan for responses or adaptive responses to disturbances. Thus, we have continued to use the ecological definition of resilience (Holling, 1973, Levin and Lubchenco, 2008) rather than the broader, socio-ecological perspective of plan, absorb, recover and adapt (doi: 10.1038/nclimate2227) that this reviewer suggest we adopt.

We provide here a brief comparison of these two resilience definitions in more detail to better understand why they are different. To begin with, both use the concept of a system trajectory, such as a population trajectory, and how it changes with a disturbance (as illustrated in Fig 1 of doi: 10.1038/nclimate2227). A realisation of this is provided in the DAG (multi-level directed acyclic graphs) model in (doi: 10.1038/srep19540). In equation 4 (doi: 10.1038/srep19540), we note that resilience is defined as a ratio of two system trajectories: (i) the critical functionality $K(t)$ for a given scenario E summed over time, and (ii) baseline functionality $K^{\text{nominal}}(t)$ summed over time. In our paper, we similarly compare the population trajectory $\mu(t)$ (Methods: Dredging) of a given scenario to the baseline population trajectory (Main Text, starting third last paragraph).

However, despite the similarities in the use of scenario trajectories and baseline trajectories, the key difference resides in how resilience is formulated from these trajectories. In (doi: 10.1038/srep19540), resilience is specified by a **single metric** that integrates (sum over time and scenarios E) into all the stages of plan, absorb, recover and adapt (Equation 4, doi: 10.1038/srep19540). However, we have adopted the approach of (Holling, 1973) and (Levin and Lubchenco, 2008) where resilience is specified by a **collection of metrics**. This vector contains metrics characterising the differences between the scenario trajectory and baseline trajectory which relate to factors such as resistance, recovery time and persistence. Although there are advantages to using a single metric (e.g. ease of interpretation), we have adopted the composite metric as: (i) it supports multi-criteria interpretation from an ecological perspective as befits our goal of understanding ecological windows, and (ii) given the wide adoption of Holling's as well as Levin and Lubchenco's approach in studies of ecological resilience..

In addition, there is a focus on socio-technical or socio-ecological systems inherent in the definition of plan, absorb, recover and adapt (doi: 10.1038/nclimate2227, doi: 10.1038/srep19540) and this approach has attracted some criticism from others studying socio-ecological processes as it assumes

that the social and biophysical world can be best analysed using unifying concepts and the same methods (Castree et al., 2014). Although we incorporate some social element in terms of dredging designs in our study, we have focused on the seagrass ecosystem as our aim was to better understand ecological windows in the biophysical world and how disturbances might affect ecosystems, not specifically how to plan for responses or adaptive responses to them.

Castree, Noel, et al. "Changing the intellectual climate." *Nature climate change* 4.9 (2014): 763-768.

To help clarify our focus, we have added:

(1) We have re-structured the introductory paragraphs to clarify our focus on stressors, definitions of resilience, ecological windows, our approach, and its application to the seagrass dredging problem. Included in this is an acknowledgement of other definitions of resilience including the plan, absorb, recover, adapt framework. Additionally, we have clarified our focus on ecological resilience in understanding ecological windows as a tool to assist managers. For an extract, refer to the response to the editor above.

(2) Clarified notation to indicate we are studying trajectories (Methods: Dredging):

For the purposes of assessing deviations from the baseline, we propose a weighted mean approach to aggregate multiple state probability trajectories into a single trajectory for comparison; this weighted mean $\mu(t)$ trajectory over time t is calculated as follows:

$$\mu(t) = p_0(t)\mathbb{I}_{\mu'(t)=0} + (1 - p_0(t))\mu'(t)\mathbb{I}_{\mu'(t)>0}$$

where $p_0(t)$ is the posterior probability of being in a zero state (if such a state exists – e.g. zero shoot density), the weighted mean $\mu'(t) = \sum_{j \in \text{states}} p_j(t)\bar{x}_j$ where $p_j(t)$ is the posterior probability of being in state j and \bar{x}_j is the mean or quantile value for state j – this value is derived from state thresholds. For example, given thresholds of 81-100% for high shoot density, the quantiles are $\bar{x}_j = \{81, 86, 91, 96, 100\}$ assuming a uniform distribution. Here we used the median, which equals the mean for a uniform distribution, for recovery and resistance calculations. \mathbb{I} is an indicator function such that: $\mathbb{I} = \begin{cases} 0, & \mu'(t) < \alpha \\ 1, & \mu'(t) \geq \alpha \end{cases}$ where α is a threshold for being in the zero state.

(3) We have identified integration with socio-ecological modelling for practical application of plan, absorb, recover and adapt (Discussion, second last paragraph):

Further integration with other socio-ecological models could support resilience-based management such as 'plan, absorb, recover and adapt' schemes¹⁸ or multi-criteria decision frameworks to make trade-offs against cost for example {Suedel, 2008 #638}, or other such approaches²⁴.

In the abstract, we have clarified our focus on ecological resilience:

Specifically, we search for ecological windows, times during which stressor impacts on ecological resilience, defined here as risk, recovery and resistance, are minimised.

Moreover, the paper keeps insisting on superiority of their approach without discussing limitation and are using baseless statements like : "Our results contrast with current approaches that focus on a single ecological event without consideration of resilience and interacting factors." Current dredging windows are developed based on extensive consultations with multiple stakeholders integrating ecological modeling and decision analytical tools. Of course the paper presents potentially interesting addition to the relevant science, but it should be placed in appropriate way in the body of current literature and practice. By the way, of course site-specific information is carefully looked at in setting dredging windows. Please review multiple Technical Notes posted at the USACE ERDC web site on the relevant issues.

RESPONSE 2:

It is certainly not our intention to degrade existing approaches. We have clarified our definition of ecological windows and the scope of our contribution (Main Text, paragraph 3 and 4, for an extract refer to the response to the editor):

In the abstract, we have clarified:

Our results contrast with previous approaches using ecological windows focused on a single ecological event without consideration of resilience and interacting factors.

Thank you for your comments and guidance to help improve our paper.

Reviewer #2 (Remarks to the Author):

Dear authors,

Thank you for the revised version of the paper. I am satisfied with the answers and amendments done to the paper. And I recommend this for publication given that you address some minor further points:

1. The first part of the main text, page 1, has no flow. It is written in telegraphic style and reads badly. This can be easily changed.

RESPONSE 3:

We have revised (Main Text paragraph 1):

Anthropogenic stressors are causing increasing degradation of valuable marine ecosystems worldwide¹⁻³. The impacts from these stressors are exacerbated in coastal areas where coastal development and ecosystems intersect⁴. However, currently available approaches for mitigating the effects of these stressors are often limited^{5,6}. This situation is concerning as corals⁷, seagrasses^{8,9} and mangroves¹⁰ are all rapidly declining globally, at least in part due to stressors associated with water quality. The management of the effects of these stressors is further complicated by the complexity of

natural environmental stressors and disturbances and their interactions with myriad ecological processes. Such complex interactions can produce non-linear, additive and synergistic cumulative responses¹¹.

2. I think that the links you do to “Panarchy” are still unclear (Lines 68-70)

RESPONSE 4:

We have clarified (Main Text paragraph 3):

Although we have adopted Holling’s seminal definition of resilience^{13,14}, other definitions could also be applied. For instance, our generative and modular modelling approach could be adapted and/or integrated with other models for the larger scale analysis of nested adaptive cycles (i.e. panarchy¹⁷) over longer time scales.

3. While you explain the difference between environmental and ecological windows, you do not address the issue of windows of opportunities in relation to ecological windows. This must be done. The reason is that this paper is about resilience and in the resilience literature windows of opportunity are broadly used. So the new “ecological windows” have to be situated in this context also.

RESPONSE 5:

We have reviewed the suggested papers in comment 8 and incorporated a definition of ecological windows as compared to environmental windows and ‘windows of opportunity’ (Main Text, paragraph 3):

To understand resilience and its management using ecological windows, we developed a whole-of-system Dynamic Bayesian Network model. Ecological windows are periods planned in advance during which a specific stressor can occur with minimal impact on resilience¹⁷. Windows are often used to manage anthropogenic activities such as dredging¹⁸. We have adopted a modelling approach here because it would be impossible to estimate impact experimentally with sufficient certainty given the complex, variable and uncertain nature of ecosystems. The ecological windows we explore here also differ from environmental windows detailed in existing regulatory frameworks (e.g., U.S.A. National Environmental Policy Act 1969) which typically do not consider site-specific biological, environmental and stressor interactions¹⁸. Our DBN presents an opportunity to predict the emergent response and resilience of a system given temporal variation in baseline environmental and biological conditions and their interaction with different timing, duration and magnitude of stressors such as dredging. Ecological windows also differ from windows of opportunity^{19,20}, which have a broader socio-ecological focus where typically unplanned events can trigger opportunities for wide-scale institutional and ecological changes. Although we have adopted Holling’s seminal definition of resilience^{11,12}, other definitions could also be adopted in a similar context. For instance, our generative and modular modelling approach could be adapted and/or integrated with other models for the larger scale analysis of nested adaptive cycles (i.e. panarchy²¹) over longer time scales. Similarly, ways of managing resilience, such as the ‘plan, absorb, recover and adapt’ framework^{22,23} could be explored based on the outputs of our modelling approach.

We also discuss how our work relates to ‘windows of opportunity’ (Discussion, third last paragraph):

In addition, our predictive approach could be applied to the analysis of “windows of opportunity”^{19,20} and especially those that arise due to a pulse disturbance like dredging. Although windows of opportunity are often defined in a socio-ecological context, an ecological window corresponding to a scheduled disturbance could be conceptualised as a type of traction opportunity²⁰ (e.g. a window that allows for recovery). However, here, the opportunity is to better understand resilience rather than the probability of breaking out of an undesirable stable state⁵⁶.

4. Line 79 please change flushing for water circulation

Done.

5. Lines 100-101 repetitive

Removed.

6. Line 109, about *Posidonia* is this really global? I don't think so, it's mainly from Australia and the Mediterranean. Please check.

RESPONSE 6:

We agree and have clarified (Main Text paragraph 5):

Halophila and *Zostera* have global distributions in tropical and temperate climates, and *Amphibolis* is a temperate Australian endemic with a similar life history and meadow characteristics to other persistent genera that are more widely distributed (e.g. *Posidonia*).

7. Lines 553-554 this is a strong statement without any evidence, what are the references for this? Either you provide them or remove.

RESPONSE 7:

We have re-worded that sentence to clarify the scope of that statement (Methods: Factor Discretisation, last paragraph):

Another advantage of using benthic light at the meadow is that it can be directly measured and encapsulates the overall impact on light due to local hydrodynamic and weather patterns.

8. There are several papers that you can use to address the links between your ecological windows and “windows of opportunities” in the resilience literature. I recommend you to look at:
Nyström et al. 2012. In Ecosystems. This is about marine systems and should be very helpful.
Moon et al. 2014. Conservation Ecology. This can be useful as it addresses the know GBR case in Australia.

Other more general and central in the resilience literature are:

Folke et al. 2010. Ecology and Society.

Olsson P. et al. 2006. Ecology and Society.

Young O. 2010. Global Environmental Change.

Please refer to Response 5.

Thank you for an excellent and interesting work!

Thank you for your comments and guidance to help improve our paper!

Reviewer #3 (Remarks to the Author):

I am most satisfied in the authors' response to my review to focus the "ecological windows" concept and writing in this manuscript. Providing the coral reef example was illuminating and I believe will be helpful for the reader. The authors also expanded their discussion to couch the approach in Dynamic Ocean Management and to also include a more sophisticated discussion of restoration trajectories.

I appreciate the sensitivity analysis outlined in the new manuscript, it addresses some of my concerns. The responses to Reviewer #2's more specific comments regarding the expert opinion work was also helpful. However, even with the stated assumptions and the sensitivity analysis it is still not transparent to me where the "tender" points are in the model. In most modeling papers those are more clear to me than what is presented here - perhaps because I am more accustomed to looking at frequentist-based statistical models, or system dynamics simulations that use series of ODEs. I suspect, but am not sure, that the assignment of the conditional probabilities is the area with the most uncertainty? Having a statement in the revised manuscript identifying areas that are worthy of next generation investigation and improvement for this approach - or even broadly identifying areas where this modeling approach has its challenges, will be very useful to an audience introduced to these methods via this manuscript.

RESPONSE 8:

The elicitation of conditional probabilities is indeed the most challenging part of building this type of model. We have incorporated a statement to this effect (Methods: Elicitation of Probabilities):

Nevertheless, elicitation of probabilities is a challenging task due to the number of probabilities needed to parametrise the model and the innate difficulty for human experts to estimate them precisely⁵⁰. Elicitation of probabilities and integration of these with probabilities estimated from data are key areas for future work.

Thank you for your comments and guidance to help improve our paper!

REVIEWERS' COMMENTS:

Reviewer #3 (Remarks to the Author):

I feel that the response of the authors to issues raised relevant to resilience were adequate. The resilience approach used comes from Holling and Levin's work. The only recommendation I might make is to contrast this with alternative resilience frameworks so that this context is crystal clear to the readers. There is, unfortunately, much confusion on the way that this theory is applied, and clarity helps in placing work in its proper context. The authors provide one statement regarding engineering resilience and how that differs from this approach. Perhaps emphasizing that the resilience they refer to is more commonly applied to ecological research and perspectives might be useful.

Reviewer's Comments

Reviewer #3 (Remarks to the Author):

I feel that the response of the authors to issues raised relevant to resilience were adequate. The resilience approach used comes from Holling and Levin's work. The only recommendation I might make is to contrast this with alternative resilience frameworks so that this context is crystal clear to the readers. There is, unfortunately, much confusion on the way that this theory is applied, and clarity helps in placing work in its proper context. The authors provide one statement regarding engineering resilience and how that differs from this approach. Perhaps emphasizing that the resilience they refer to is more commonly applied to ecological research and perspectives might be useful.

Thank you. We have added a sentence as suggested about the use of this definition in ecology (Introduction, paragraph 2):

...We focus on ecological resilience¹² as first proposed by Holling¹¹ rather than engineering resilience which focuses exclusively on recovery¹². The former is a broader definition centred around the set of processes and structures describing an ecosystem and is widely applied in ecology¹²